# Learning place cells, grid cells and invariances with excitatory and inhibitory plasticity

**Simon Nikolaus Weber\*, Henning Sprekeler\***

Modelling of Cognitive Processes, Institute of Software Engineering and Theoretical Computer Science, Technische Universität Berlin, Berlin, Germany

**Abstract** Neurons in the hippocampus and adjacent brain areas show a large diversity in their tuning to location and head direction, and the underlying circuit mechanisms are not yet resolved. In particular, it is unclear why certain cell types are selective to one spatial variable, but invariant to another. For example, place cells are typically invariant to head direction. We propose that all observed spatial tuning patterns – in both their selectivity and their invariance – arise from the same mechanism: Excitatory and inhibitory synaptic plasticity driven by the spatial tuning statistics of synaptic inputs. Using simulations and a mathematical analysis, we show that combined excitatory and inhibitory plasticity can lead to localized, grid-like or invariant activity. Combinations of different input statistics along different spatial dimensions reproduce all major spatial tuning patterns observed in rodents. Our proposed model is robust to changes in parameters, develops patterns on behavioral timescales and makes distinctive experimental predictions.
DOI: https://doi.org/10.7554/eLife.34560.001

## Introduction

Neurons in the hippocampus and the adjacent regions exhibit a broad variety of spatial activation patterns that are tuned to position, head direction or both. Common observations in these spatial dimensions are localized, bell-shaped tuning curves (*O'Keefe, 1976*; *Taube et al., 1990*), periodically repeating activity (*Fyhn et al., 2004*; *Hafting et al., 2005*) and invariances (*Muller et al., 1994*; *Burgess et al., 2005*), as well as combinations of these along different spatial dimensions (*Sargolini et al., 2006a*; *Krupic et al., 2012*). For example, head direction cells are often invariant to location (*Burgess et al., 2005*), and place cells are commonly invariant to head direction (*Muller et al., 1994*). The cellular and network mechanisms that give rise to each of these firing patterns are subject to extensive experimental and theoretical research. Several computational models have been suggested to explain the emergence of grid cells (*Fuhs and Touretzky, 2006*; *McNaughton et al., 2006*; *Franzius et al., 2007a*; *Burak and Fiete, 2009*; *Couey et al., 2013*; *Burgess et al., 2007*; *Kropff and Treves, 2008*; *Bush and Burgess, 2014*; *Castro and Aguiar, 2014*; *Dordek et al., 2016*; *Stepanyuk, 2015*; *Giocomo et al., 2011*; *Zilli, 2012*; *D'Albis and Kempter, 2017*; *Monsalve-Mercado and Leibold, 2017*), place cells (*Tsodyks and Sejnowski, 1995*; *Battaglia and Treves, 1998*; *Arleo and Gerstner, 2000*; *Solstad et al., 2006*; *Franzius et al., 2007b*; *Burgess and O'Keefe, 2011*; *Franzius et al., 2007a*) and head direction cells (*McNaughton et al., 1991*; *Redish et al., 1996*; *Zhang, 1996*; *Franzius et al., 2007a*). Most of these models are designed to explain the spatial selectivity of one particular cell type and do not consider invariances along other dimensions, although the formation of invariant representations is a non-trivial problem (*DiCarlo and Cox, 2007*). In view of the variety of spatial tuning patterns, the question arises of whether differences in tuning of different cells in different areas reflect differences in microcircuit connectivity, single cell properties or plasticity rules, or whether there is a unifying

**\*For correspondence:**
weber@tu-berlin.de (SNW);
h.sprekeler@tu-berlin.de (HS)

**Competing interests:** The authors declare that no competing interests exist.

**eLife digest** Knowing where you are never hurts, be it during a holiday in New York or on a hiking trip in the Alps. Our sense of location seems to depend on a structure deep within the brain called the hippocampus, and its neighbor, the entorhinal cortex. Studies in rodents have shown that these areas act a little like an in-built GPS for the brain. They contain different types of neurons that help the animal to work out where it is and where it is going. Among those are place cells, present within the hippocampus, and grid cells and head direction cells, found within the entorhinal cortex and other areas.

Place cells fire whenever an animal occupies a specific location in its environment, with each place cell firing at a different spot. Grid cells generate virtual maps of the surroundings that resemble grids of repeating triangles. Whenever an animal steps onto a corner of one of these virtual triangles, the grid cell that generated that map starts to fire. Head direction cells increase their firing whenever an animal's head is pointing in a specific direction. These cell types thus provide animals with complementary information about their location. But how do the cells first become selective for specific places or head directions?

Weber and Sprekeler propose that a single mechanism gives rise to the spatial characteristics of all these different types of cells. Like all neurons, these cells communicate with their neighbors at junctions called synapses. These may be either excitatory or inhibitory. Cells at excitatory synapses activate their neighbors, whereas cells at inhibitory synapses deactivate them. Weber and Sprekeler used a computer to simulate changes in excitatory and inhibitory synapses in a virtual rat exploring an environment. Interactions between the two types of synapses gave rise to virtual cells that behaved like place, grid or head direction cells. Which cell type emerged depended on whether the excitatory or the inhibitory synapses were more sensitive to the virtual rat's location.

This idea adds to a range of others proposed to explain how the brain codes for locations. Whether any of these ideas or a combination of them is correct remains to be determined. Further pieces are needed if we are to solve the puzzle of how the brain supports navigation.
DOI: https://doi.org/10.7554/eLife.34560.002

principle. In this paper we suggest that both the observed spatial selectivities and invariances can be explained by a common mechanism – interacting excitatory and inhibitory synaptic plasticity – and that the observed differences in the response profiles of grid, place and head direction cells result from differences in the spatial tuning of excitatory and inhibitory synaptic afferents. Here, we explore this hypothesis in a computational model of a feedforward network of rate-based neurons. Simulations as well as a mathematical analysis indicate that the model reproduces the large variety of response patterns of neurons in the hippocampal formation and adjacent areas and can be used to make predictions for the input statistics of each cell type.

## Results

We study the development of spatial representations in a network of rate-based neurons with interacting excitatory and inhibitory plasticity. A single model neuron that represents a cell in the hippocampal formation or adjacent areas receives feedforward input from excitatory and inhibitory synaptic afferents. As a simulated rat moves through an environment, these synaptic afferents are weakly modulated by spatial location and in later sections also by head direction. This modulation is irregular and non-localized with multiple maxima (*Buetfering et al., 2014*); see *Figure 1a* and Materials and methods. Importantly, different inputs show different modulation profiles and each profile is temporally stable. We also show results for localized, that is, place cell-like, input (*O'Keefe and Dostrovsky, 1971*; *Marshall et al., 2002*; *Wilent and Nitz, 2007*). The output rate is given by a weighted sum of the excitatory and inhibitory inputs.

In our model, both excitatory and inhibitory synaptic weights are subject to plasticity. The excitatory weights change according to a Hebbian plasticity rule (*Hebb, 1949*) that potentiates the weights in response to simultaneous pre- and postsynaptic activity. The inhibitory synapses evolve according to a plasticity rule that changes their weights in proportion to presynaptic activity and the difference between postsynaptic activity and a target rate (1 Hz in all simulations). This rule has

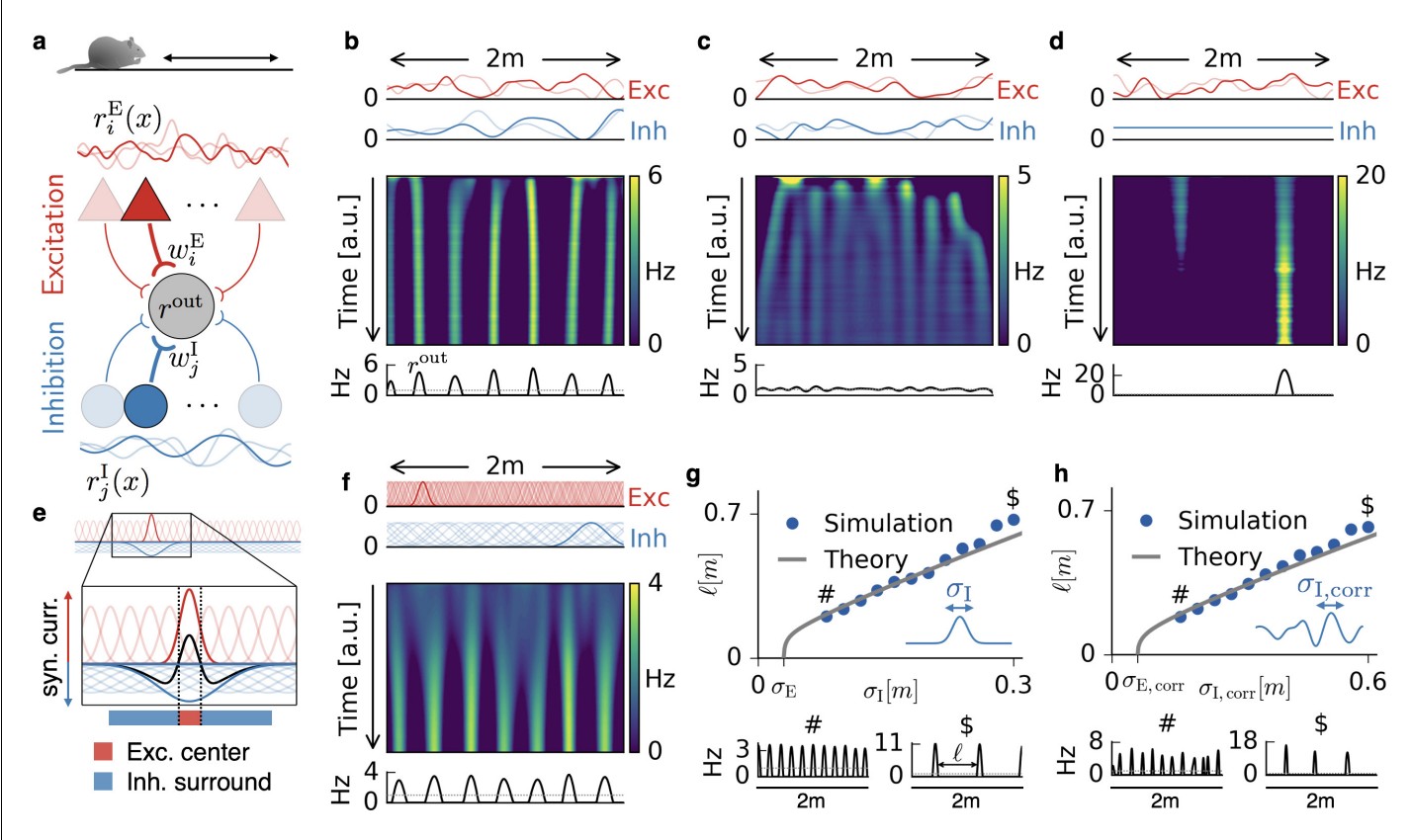

**Figure 1.** Emergence of periodic, invariant and single field firing patterns. (a) Network model for a linear track. A threshold-linear output neuron (gray) receives input from excitatory (red) and inhibitory (blue) cells, which are spatially tuned (curves on top and bottom). (b) Spatially tuned input with smoother inhibition than excitation. The fluctuating curves (top) show two exemplary spatial tunings (one is highlighted) of excitatory and inhibitory input neurons. Interacting excitatory and inhibitory synaptic plasticity gradually changes an initially random response of the output neuron (firing rate $r^{\mathrm{out}}$) into a periodic, grid cell-like activity pattern. (c) If the spatial tuning of inhibitory input neurons is less smooth than that of excitatory input neurons, the interacting excitatory and inhibitory plasticity leads to a spatially invariant firing pattern. The output neuron fires close to the target rate of 1 Hz everywhere. (d) For very smooth or spatially untuned inhibitory inputs, the output neuron develops a single firing field, reminiscent of a place cell. (e) The mechanism, illustrated for place cell-like input. When a single excitatory weight is increased relative to the others, the balancing inhibitory plasticity rule leads to an immediate increase of inhibition at the associated location. If inhibitory inputs are smoother than excitatory inputs, the resulting approximate balance creates a center surround field: a local overshoot of excitation (firing field) surrounded by an inhibitory corona. The next firing field emerges at a distance where the inhibition has faded out. Iterated, this results in a spatially periodic arrangement of firing fields. (f) Inputs with place field-like tuning. Gaussian curves (top) show the spatial tuning of excitatory and inhibitory input neurons (one neuron of each kind is highlighted, 20 percent of all inputs are displayed). A grid cell firing pattern emerges from an initially random weight configuration. (g) Grid spacing $\ell$ scales with inhibitory tuning width $\sigma_{\mathrm{I}}$. Simulation results (dots) agree with a mathematical bifurcation analysis (solid). Output firing rate examples at the two indicated locations are shown at the bottom. (h) Inhibitory smoothness $\sigma_{\mathrm{I,corr}}$ controls grid spacing; arrangement as in (d). Note that the time axes in (b, c,d,f) are different, because the speed at which the patterns emerge is determined by both the learning rates of the plasticity and the firing rate of the input neurons. We kept the learning rate constant and adjusted the simulation times to achieve convergence. Choosing identical simulation times, but different learning rates, leads to identical results (*Figure 1—figure supplement 2*). Rat clip art from [https://openclipart.org/detail/216359/klara; 2015].
DOI: https://doi.org/10.7554/eLife.34560.003

The following figure supplements are available for figure 1:

**Figure supplement 1.** Statistics of the synaptic weights.
DOI: https://doi.org/10.7554/eLife.34560.004

**Figure supplement 2.** Different learning rates lead to identical results.
DOI: https://doi.org/10.7554/eLife.34560.005

previously been shown to balance excitation and inhibition such that the firing rate of the output neuron approaches the target rate (*Vogels et al., 2011*; *D'Amour and Froemke, 2015*). We assume the inhibitory plasticity will act fast enough to track changes of excitatory weights, so that excitation and inhibition are approximately balanced at all times.

## Relative spatial smoothness of the excitatory and inhibitory input determines the firing pattern of the output neuron

We first simulate a rat that explores a linear track (*Figure 1*). The spatial tuning of each input neuron is stable in time and depends smoothly on the location of the animal, but is otherwise random (e.g. *Figure 1a*). As a measure of smoothness, we use the spatial autocorrelation length. In the following, this is the central parameter of the input statistics, which is chosen separately for excitation and inhibition. In short, we assume that temporally stable spatial information is presynaptically present but we have minimal requirements on its format, aside from the spatial autocorrelation length.

At the beginning of each simulation, all synaptic weights are random. As the animal explores the track, the excitatory and inhibitory weights change in response to pre- and postsynaptic activity, and the output cell gradually develops a spatial activity pattern. We find that this pattern is primarily determined by whether the excitatory or inhibitory inputs are smoother in space. If the inhibitory tuning is smoother than the excitatory tuning (*Figure 1b*), the output neuron develops equidistant firing fields, reminiscent of grid cells on a linear track (*Hafting et al., 2008*). If instead the excitatory tuning is smoother, the output neuron fires close to the target rate of 1 Hz everywhere (*Figure 1c*); it develops a spatial invariance. For spatially untuned inhibitory afferents (*Grienberger et al., 2017*), the output neuron develops a single firing field, reminiscent of a one-dimensional place cell (*Figure 1d*); (cf. *Clopath et al., 2016*).

The emergence of these firing patterns can be best explained in the simplified scenario of place field-like input tuning (*Figure 1e,f*). The spatial smoothness is then given by the size of the place fields. Let us assume that the output neuron fires at the target rate everywhere (see Materials and methods). From this homogeneous state, a small potentiation of one excitatory weight leads to an increased firing rate of the output neuron at the location of the associated place field (highlighted red curve in *Figure 1e*). To bring the output neuron back to the target rate, the inhibitory learning rule increases the synaptic weight of inhibitory inputs that are tuned to the same location (highlighted blue curve in *Figure 1e*). If these inhibitory inputs have smaller place fields than the excitatory inputs (*Figure 1c*), this restores the target rate everywhere (*Vogels et al., 2011*). Hence, inhibitory plasticity can stabilize spatial invariance if the inhibitory inputs are sufficiently precise (i.e. not too smooth) in space. In contrast, if the spatial tuning of the inhibitory inputs is smoother than that of the excitatory inputs, the target firing rate cannot be restored everywhere. Instead, the compensatory potentiation of inhibitory weights increases the inhibition in a spatial region at least the size of the inhibitory place fields. This leads to a corona of inhibition, in which the output neuron cannot fire (*Figure 1e*, blue region). Outside of this inhibitory surround the output neuron can fire again and the next firing field develops. Iterated, this results in a periodic arrangement of firing fields (*Figure 1f* and Figure 7b for a depiction of the input currents). Spatially untuned inhibition corresponds to a large inhibitory corona that exceeds the length of the linear track, so that only a single place field remains. From a different perspective, spatially untuned input can also be understood as a limit case of vanishing spatial variation in the firing rate rather than a limit of infinite smoothness. Consistent with this view, a development of grid patterns or invariance requires a sufficiently strong spatial modulation of the inhibitory inputs (Materials and methods).

The argument of the preceding paragraph can be extended to the scenario where input is irregularly modulated by space. For non-localized input tuning (*Figure 1b,c,d*), any weight change that increases synaptic input in one location will also increase it in a surround that is given by the smoothness of the input tuning (see Materials and methods for a mathematical analysis). In the simulations, the randomness manifests itself in occasional defects in the emerging firing pattern (*Figure 1h*, bottom, and *Figure 1—figure supplement 1*). The above reasoning suggests that the width of individual firing fields is determined by the smoothness of the excitatory input tuning, while the distance between grid fields, that is, the grid spacing, is set by the smoothness of the inhibitory input tuning. Indeed, both simulations and a mathematical analysis (Materials and methods) confirm that the grid spacing scales linearly with the inhibitory smoothness in a large range, both for localized (*Figure 1g*) and non-localized input tuning (*Figure 1h*). The analysis also reveals a weak logarithmic dependence of the grid spacing on the ratio of the learning rates, the mean firing rates and the number of afferents of the excitatory and inhibitory population (*Equation 78* and Figure 8b).

In summary, the interaction of excitatory and inhibitory plasticity can lead to spatial invariance, spatially periodic activity patterns or single place fields depending on the spatial statistics of the excitatory and inhibitory input.

## Emergence of hexagonal firing patterns

When a rat navigates in a two-dimensional arena, the spatial firing maps of grid cells in the medial entorhinal cortex (mEC) show pronounced hexagonal symmetry (*Hafting et al., 2005*; *Fyhn et al., 2004*) with different grid spacings and spatial phases. To study whether a hexagonal firing pattern can emerge from interacting excitatory and inhibitory plasticity, we simulate a rat in a quadratic arena. The rat explores the arena for 10 hr, following trajectories extracted from behavioral data (*Sargolini et al., 2006b*); Materials and methods. To investigate the role of the input statistics, we consider three different classes of input tuning: (i) place cell-like input (*Figure 2a*), (ii) sparse non-localized input, in which the tuning of each input neuron is given by the sum of 100 randomly located place fields (*Figure 2b* and (iii) dense non-localized input, in which the tuning of each input is a random function with fixed spatial smoothness (*Figure 2c*). For all input classes, the spatial tuning of the inhibitory inputs is smoother than that of the excitatory inputs.

Initially, all synaptic weights are random and the activity of the output neuron shows no spatial symmetry. While the rat forages through the environment, the output cell develops a periodic firing pattern for all three input classes, reminiscent of grid cells in the mEC (*Fyhn et al., 2004*; *Hafting et al., 2005*) and typically with the same hexagonal symmetry. This hexagonal arrangement is again a result of smoother inhibitory input tuning, which generates a spherical inhibitory corona around each firing field (compare *Figure 1e*). These center-surround fields are arranged in a hexagonal pattern – the closest packing of spheres in two dimensions; (cf. *Turing, 1952*). We find that the spacing of this pattern is determined by the inhibitory smoothness. The similarity between cells in terms of orientation and phase of the grid depends – in decreasing order – on whether they receive the same inputs, on the trajectories on which the tuning was learned and on the initial synaptic weights (*Figure 2—figure supplement 1*). Two grid cells can thus have different phase and orientation, even if they share a large fraction or all of their inputs.

For the linear track, the randomness of the non-localized inputs leads to defects in the periodicity of the grid pattern. In two dimensions, we find that the randomness leads to distortions of the hexagonal grid. To quantify this effect, we simulated 500 random trials for each of the three input scenarios and plotted the grid score histogram (Appendix 1) before and after 10 hr of spatial exploration (*Figure 2d,e,f*). Different trials have different trajectories, different initial synaptic weights and different random locations of the input place fields (for sparse input) or different random input functions (for dense input). For place cell-like input, most of the output cells develop a positive grid score during 10 hr of spatial exploration (33% before to 86% after learning, *Figure 2d*). Even for low grid scores, the firing rate maps look grid-like after learning but exhibit a distorted symmetry (*Figure 2d*). For sparse non-localized input, the fraction of output cells with a positive grid score increases from 35% to 87% and for dense non-localized input from 16% to 68% within 10 hr of spatial exploration (*Figure 2e,f*). The excitatory and inhibitory inputs are not required to have the same tuning statistics. Grid patterns also emerge when excitation is localized and inhibition is non-localized (*Figure 2—figure supplement 2*).

In summary, the interaction of excitatory and inhibitory plasticity leads to grid-like firing patterns in the output neuron for all three input scenarios. The grids are typically less distorted for sparser input (*Figure 2g*).

## Rapid appearance of grid cells and their reaction to modifications of the environment

In unfamiliar environments, neurons in the mEC exhibit grid-like firing patterns within minutes (*Hafting et al., 2005*). Moreover, grid cells react quickly to changes in the environment (*Fyhn et al., 2007*; *Savelli et al., 2008*; *Barry et al., 2012*). These observations challenge models for grid cells that require gradual synaptic changes during spatial exploration. In principle, the time scale of plasticity-based models can be augmented arbitrarily by increasing the synaptic learning rates. For stable patterns to emerge, however, significant weight changes must occur only after the animal has visited most of the environment. To explore the edge of this trade-off between speed and stability,

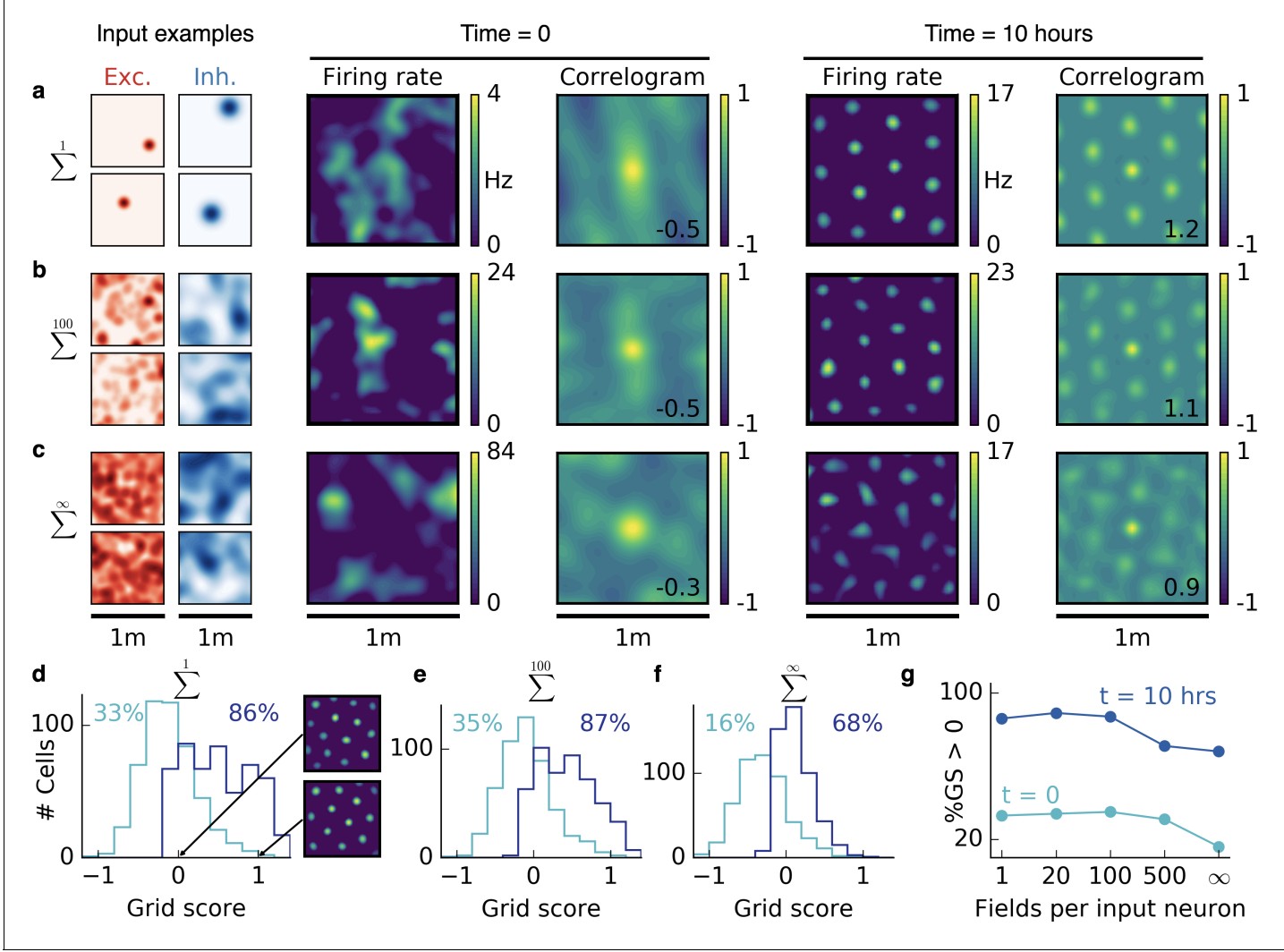

**Figure 2.** Emergence of two-dimensional grid cells. (a,b,c) Columns from left to right: Spatial tuning of excitatory and inhibitory input neurons (two examples each); spatial firing rate map of the output neuron and corresponding autocorrelogram before and after spatial exploration of 10 hr. The number on the correlogram shows the associated grid score. Different rows correspond to different spatial tuning characteristics of the excitatory and inhibitory inputs. For all figures the spatial tuning of inhibitory input neurons is smoother than that of excitatory input neurons. (a) Each input neuron is a place cell with random location. (b) The tuning of each input neuron is given as the sum of 100 randomly located place fields. (c) The tuning of each input neuron is a random smooth function of the location. This corresponds to the sum of infinitely many randomly located place fields. Before learning, the spatial tuning of the output neuron shows no symmetry. After 10 hr of spatial exploration the output neuron developed a hexagonal pattern. (d) Grid score histogram for 500 output cells with place cell-like input. Before learning (light blue), 33% of the output cells have a positive grid score. After 10 hr of spatial exploration (dark blue), this value increases to 86%. Two example rate maps are shown. The arrows point to the grid score of the associated rate map. Even for low grid scores the learned firing pattern looks grid-like. (e,f) Grid score histograms for input tuning as in (b,c), arranged as in (d). (g) Fraction of neurons with positive grid score before (light blue) and after learning (dark blue) as a function of the number of fields per input neuron. Note that to learn within 10 hr of exploration time, we used different learning rates for different input scenarios. Using identical learning rates for all input scenarios but adjusting the simulation times to achieve convergence leads to identical results (*Figure 2—figure supplement 6*).

DOI: https://doi.org/10.7554/eLife.34560.006

The following figure supplements are available for figure 2:

**Figure supplement 1.** Influence of random simulation parameters on the final grid pattern.

DOI: https://doi.org/10.7554/eLife.34560.007

**Figure supplement 2.** Using different input statistics for different populations also leads to hexagonal firing patterns.

DOI: https://doi.org/10.7554/eLife.34560.008

**Figure supplement 3.** Boundary effects in simulations with place field-like input.

DOI: https://doi.org/10.7554/eLife.34560.009

**Figure supplement 4.** Weight normalization is not crucial for the emergence of grid cells.

*Figure 2 continued on next page*

*Figure 2 continued*

DOI: https://doi.org/10.7554/eLife.34560.010

**Figure supplement 5.** Distribution of input fields.

DOI: https://doi.org/10.7554/eLife.34560.011

**Figure supplement 6.** Different learning rates lead to identical results.

DOI: https://doi.org/10.7554/eLife.34560.012

we increased the learning rates to a point where the grids are still stable but where further increase would reduce the stability (*Figure 3—figure supplement 1*). For place cell-like input, periodic patterns can be discerned within 10 min of spatial exploration, starting with random initial weights (*Figure 3a,b*). The pattern further emphasizes over time and remains stable for many hours (*Figure 3c* and *Figure 3—figure supplement 2*).

To investigate the robustness of this phenomenon, we ran 500 realizations with different trajectories, initial synaptic weights and locations of input place fields. In all simulations, a periodic pattern emerged within the first 30 min, and a majority of patterns exhibited hexagonal symmetry after 3 hr (increasing from 33% to 81%, *Figure 3c,d*). For non-localized input, the emergence of the final grids typically takes longer, but the first grid fields are also observed within minutes and are still present in the final grid, as observed in experiments (*Hafting et al., 2005*); (*Figure 3—figure supplement 3*).

Above, we modeled the exploration of a previously unknown room by assuming the initial synaptic weights to be randomly distributed. If the rat had previous exposure to the room or to a similar room, a structure might already have formed in some of the synaptic weights. This structure could aid the development of the grid in similar rooms or hinder it in a novel room. To study this, we simulate a network that first learns the synaptic weights in one room. We then introduce a graded modification of the room by remapping the firing fields of a fraction of input neurons to random locations. We find that the output firing pattern is robust to such perturbations, even if more than half of the inputs are remapped (*Figure 3—figure supplement 2*). If all inputs are changed, corresponding to a novel room, a grid pattern is learned anew. The strong initial pattern in the weights does not hinder this development (*Figure 3—figure supplement 2*).

Recently, *Wernle et al., 2018* discovered that in an arena separated by a wall, single grid cells form two independent grid patterns — one on each side of the wall — that coalesce once the wall is removed. They find that grid fields close to the partition wall move to establish a more coherent pattern. In contrast, fields far away from the partition wall do not change their locations. Rosay et al. reproduced this experimental finding by simulating grid fields as interacting particles (Rosay et al., in preparation). They also demonstrated how it could be reproduced by a feedforward model for grid cells based on firing rate adaptation (Rosay et al., in preparation; *Kropff and Treves, 2008*). Inspired by these experiments and simulations, we simulate a rat that first explores one half of a quadratic arena and then the other half, for 2.5 hr each (*Figure 4a*). A grid pattern emerges in each compartment (*Figure 4b,c*). We then remove the partition wall and the rat explores the entire arena for another 5 hr (*Figure 4a*). As observed experimentally, grid fields close to the former partition line rearrange to make the two grids more coherent and grid fields far away from the partition line basically stay where they were (*Figure 4d*).

In summary, periodic patterns emerge rapidly in our model and the associated time scale is limited primarily by how quickly the animal visits its surroundings, that is, by the same time scale that limits the experimental recognition of the grids.

## Place cells, band cells and stretched grids

In addition to grids, the mEC and adjacent brain areas exhibit a plethora of other spatial activity patterns including spatially invariant (*Burgess et al., 2005*), band-like (periodic along one direction and invariant along the other) (*Krupic et al., 2012*), and spatially periodic but non-hexagonal patterns (*Krupic et al., 2012*; *Hardcastle et al., 2017*; *Diehl et al., 2017*). Note that it is currently debated whether or not some of the observed spatially periodic but non-hexagonal firing patterns are artifacts of poorly isolated single cell data in multi-electrode recordings (*Navratilova et al., 2016*; *Krupic et al., 2015b*). In contrast to spatially periodic tuning, place cells in the hippocampus proper are typically only tuned to a single or few locations in a given environment (*O'Keefe and*

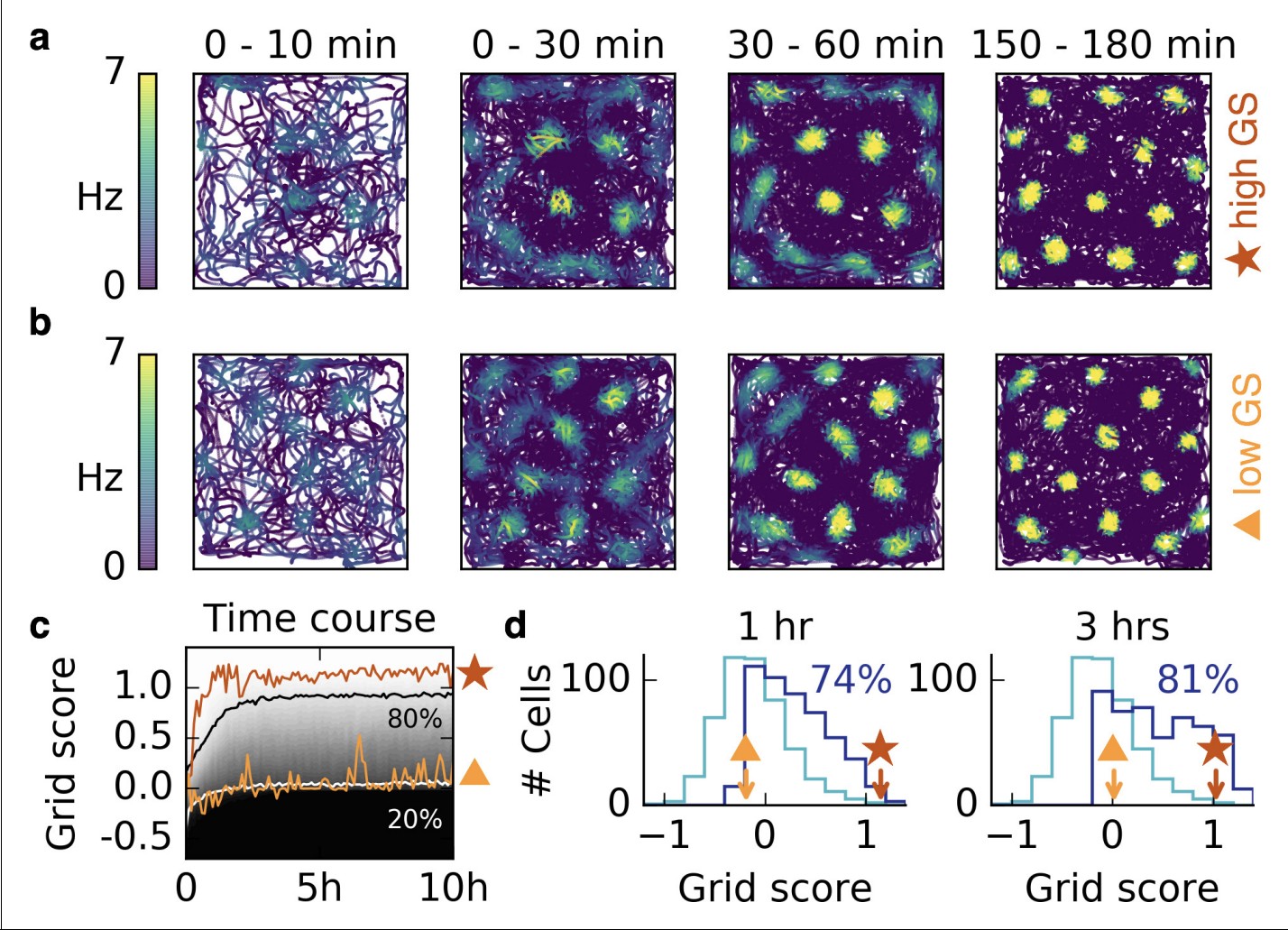

**Figure 3.** Grid patterns form rapidly during exploration and remain stable for many hours. (a,b) Rat trajectories with color-coded firing rate of a cell that receives place cell-like input. The color depicts the firing rate at the time of the location visit, not after learning. Bright colors indicate higher firing rates. The time interval of the trajectory is shown above each plot. Initially all synaptic weights are set to random values. Parts (a) and (b) show two different realizations with a good (red star) and a bad (orange triangle) grid score development. After a few minutes a periodic structure becomes visible and enhances over time. (c) Time course of the grid score in the simulations shown in (a) (red) and (b) (orange). While the periodic patterns emerge within minutes, the manifestation of the final hexagonal pattern typically takes a couple of hours. Once the pattern is established it remains stable for many hours. The gray scale shows the cumulative histogram of the grid scores of 500 realizations (black = 0, white = 1). The solid white and black lines indicate the 20% and 80% percentiles, respectively. (d) Histogram of grid scores of the 500 simulations shown in (c). Initial histogram in light blue, histogram after 1 hr and after 3 hr in dark blue. Numbers show the fraction of cells with positive grid score at the given time. Rat trajectories taken from *Sargolini et al., 2006b*).

DOI: https://doi.org/10.7554/eLife.34560.013

The following figure supplements are available for figure 3:

**Figure supplement 1.** Learning too fast leads to unstable grids.

DOI: https://doi.org/10.7554/eLife.34560.014

**Figure supplement 2.** Influence of input remapping on grid patterns.

DOI: https://doi.org/10.7554/eLife.34560.015

**Figure supplement 3.** Rapid development of grid patterns from non-localized input.

DOI: https://doi.org/10.7554/eLife.34560.016

*Dostrovsky, 1971*; *Moser et al., 2008*; *Leutgeb et al., 2005*). If the animal traversed the environment along a straight line, all of these cells would be classified as periodic, localized or invariant (*Figure 1*), although the classification could vary depending on the direction of the line. Based on this

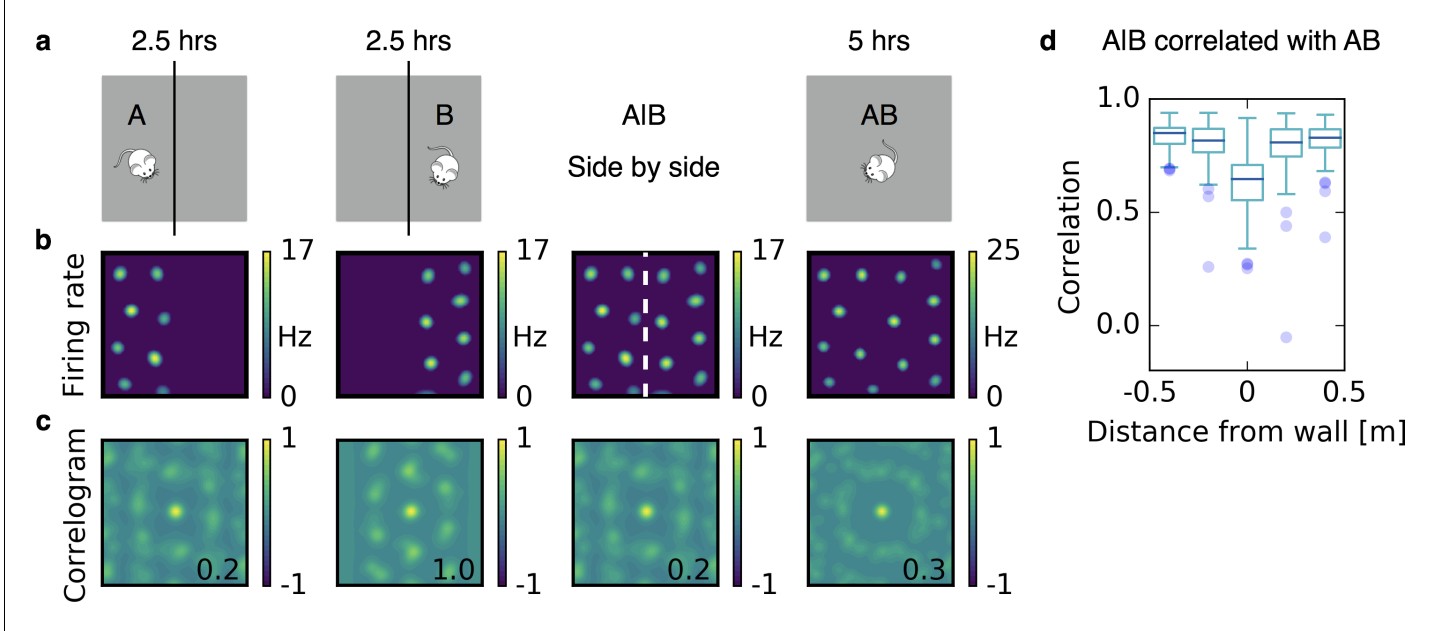

**Figure 4.** Grids coalesce in contiguous environments. (a) Illustration of the experiment. A quadratic arena (gray box) is divided into two rectangular compartments by a wall (black line). The animal explores one compartment (A) and then the other compartment (B) for 2.5 hr each. Then the wall is removed and the rat explores the entire arena (AB) for 5 hr. (b) Firing rate maps. From left to right: After learning in A; after learning in B; the maps from A and B shown side by side (A|B); after learning in AB. (c) Autocorrelograms of the rate maps shown in (b). The number inside the correlogram shows the grid score. (d) Box plot of the correlations of the firing rate map A|B with the firing rate map AB as a function of distance from the partition wall. Close to the partition wall the correlation is low, far away from the partition wall it is high. This indicates that grid fields rearrange only locally. Each box extends from the first to the third quartile, with a dark blue line at the median. The lower whisker reaches from the lowest data point still within 1.5 IQR of the lower quartile, and the upper whisker reaches to the highest data point still within 1.5 IQR of the upper quartile, where IQR is the inter quartile range between the third and first quartile. Dots show flier points. Data: 100 realizations of experiments as in (a,b,c). For simulation details see Appendix 1. Mouse clip art from lemmling, https://openclipart.org/detail/17622/simple-cartoon-mouse-1; 2006.
DOI: https://doi.org/10.7554/eLife.34560.017

observation, we hypothesized that all of these patterns could be the result of an input autocorrelation structure that differs along different spatial directions.

We first verified that also in a two-dimensional arena, place cells emerge from a very smooth inhibitory input tuning (*Figure 5a,b*). The emergence of place cells is independent of the exact shape of the excitatory input. Non-localized inputs (*Figure 5a*) lead to similar results as those from grid cell-like inputs of different orientation and grid spacing (*Figure 5b*, Methods and materials); for other models for the emergence of place cells from grid cells see (*Solstad et al., 2006*; *Franzius et al., 2007b*; *Rolls et al., 2006*; *Molter and Yamaguchi, 2008*; *Ujfalussy et al., 2009*; *Savelli and Knierim, 2010*). Next we verified that also in two dimensions, spatial invariance results when excitation is broader than inhibition (*Figure 5c*). We then varied the smoothness of the inhibitory inputs independently along two spatial directions. If the spatial tuning of inhibitory inputs is smoother than the tuning of the excitatory inputs along one dimension but less smooth along the other, the output neuron develops band cell-like firing patterns (*Figure 5d*). If inhibitory input is smoother than excitatory input, but not isotropic, the output cell develops stretched grids with different spacing along two axes (*Figure 5e*). For these anisotropic cases, stretched hexagonal grids and rectangular arrangements of firing fields appear similarly favorable (compare *Figure 5e*, second row and column). A hexagonal arrangement is favored by a dense packing of inhibitory coronas, whereas a rectangular arrangement would maximize the proximity of the excitatory centers, given the inhibitory corona (*Figure 5—figure supplement 1*).

In summary, the relative spatial smoothness of inhibitory and excitatory input determines the symmetry of the spatial firing pattern of the output neuron. The requirements for the input tuning that support invariance, periodicity and localization apply individually to each spatial dimension, opening up a combinatorial variety of spatial tuning patterns.

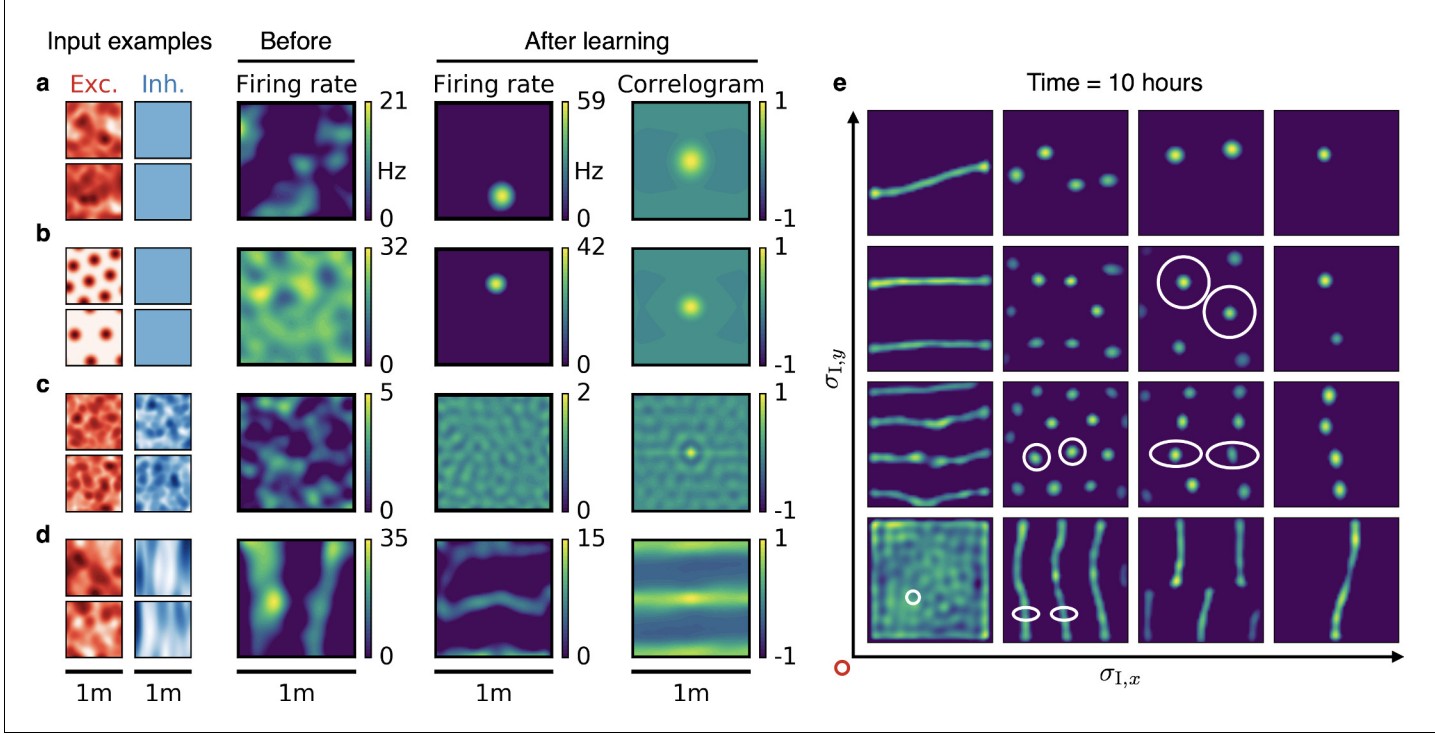

**Figure 5.** Emergence of spatially tuned cells of diverse symmetries. (a,b,c,d) Arrangement as in *Figure 2*. (a,b) Place cells emerge if the inhibitory autocorrelation length exceeds the box length or if the inhibitory neurons are spatially untuned. The type of tuning of the excitatory input is not crucial: Place cells develop for non-localized input (a) as well as for grid cell input (b). (c) The output neuron develops an invariance if the spatial tuning of inhibitory input neurons is less smooth than the tuning of excitatory input neurons. (d) Band cells emerge if the spatial tuning of inhibitory input is asymmetric, such that its autocorrelation length is larger than that of excitatory input along one direction (here the *y*-direction) and smaller along the other (here the *x*-direction). (e) Overview of how the shape of the inhibitory input tuning determines the firing pattern of the output neuron. Each element depicts the firing rate map of the output neuron after 10 hr. White ellipses of width $2\sigma_{I,x}$ and $2\sigma_{I,y}$ in $x-$ and $y-$direction indicate the direction-dependent standard deviation of the spatial tuning of the inhibitory input neurons. For simplicity, the width of the excitatory tuning fields, $\sigma_E$, is the same in all simulations. It determines the size of the circular firing fields. The red circle at the axis origin is of diameter $2\sigma_E$.

DOI: https://doi.org/10.7554/eLife.34560.018

The following figure supplement is available for figure 5:

**Figure supplement 1.** Arrangement of firing fields for asymmetric input.

DOI: https://doi.org/10.7554/eLife.34560.019

## Spatially tuned input combined with head direction selectivity leads to grid, conjunctive and head direction cells

Many cells in and around the hippocampus are tuned to the head direction of the animal (*Taube et al., 1990*; *Taube, 1995*; *Chen et al., 1994*). These head direction cells are typically tuned to a single head direction, just like place cells are typically tuned to a single location. Moreover, head direction cells are often invariant to location (*Burgess et al., 2005*), just like place cells are commonly invariant to head direction (*Muller et al., 1994*). There are also cell types with conjoined spatial and head direction tuning. Conjunctive cells in the mEC fire like grid cells in space, but only in a particular head direction (*Sargolini et al., 2006a*), and many place cells in the hippocampus of crawling bats also exhibit head direction tuning (*Rubin et al., 2014*). To investigate whether these tuning properties could also result in our model, we simulated a rat that moves in a square box, whose head direction is constrained by the direction of motion (Appendix 1). Each input neuron is tuned to both space and head direction (see *Figure 6* for localized and *Figure 6—figure supplement 1* for non-localized input).

In line with the previous observations, we find that the spatial tuning of the output neuron is determined by the relative spatial smoothness of the excitatory and inhibitory inputs, and the head direction tuning of the output neuron is determined by the relative smoothness of the head direction

tuning of the inputs from the two populations. If the head direction tuning of excitatory input neurons is smoother than that of inhibitory input neurons, the output neuron becomes invariant to head direction (*Figure 6a*). If instead only the excitatory input is tuned to head direction, the output neuron develops a single activity bump at a particular head direction (*Figure 6b,c*). The concurrent spatial tuning of the inhibitory input neurons determines the spatial tuning of the output neuron. For spatially smooth inhibitory input, the output neuron develops a hexagonal firing pattern (*Figure 6a, b*), and for less smooth inhibitory input the firing of the output neuron is invariant to the location of the animal (*Figure 6c*).

In summary, the relative smoothness of inhibitory and excitatory input neurons in space and in head direction determines whether the output cell fires like a pure grid cell, a conjunctive cell or a pure head direction cell (*Figure 6d*).

We find that the overall head direction tuning of conjunctive cells is broader than that of individual grid fields (*Figure 6e*). This results from variations in the preferred head direction of different

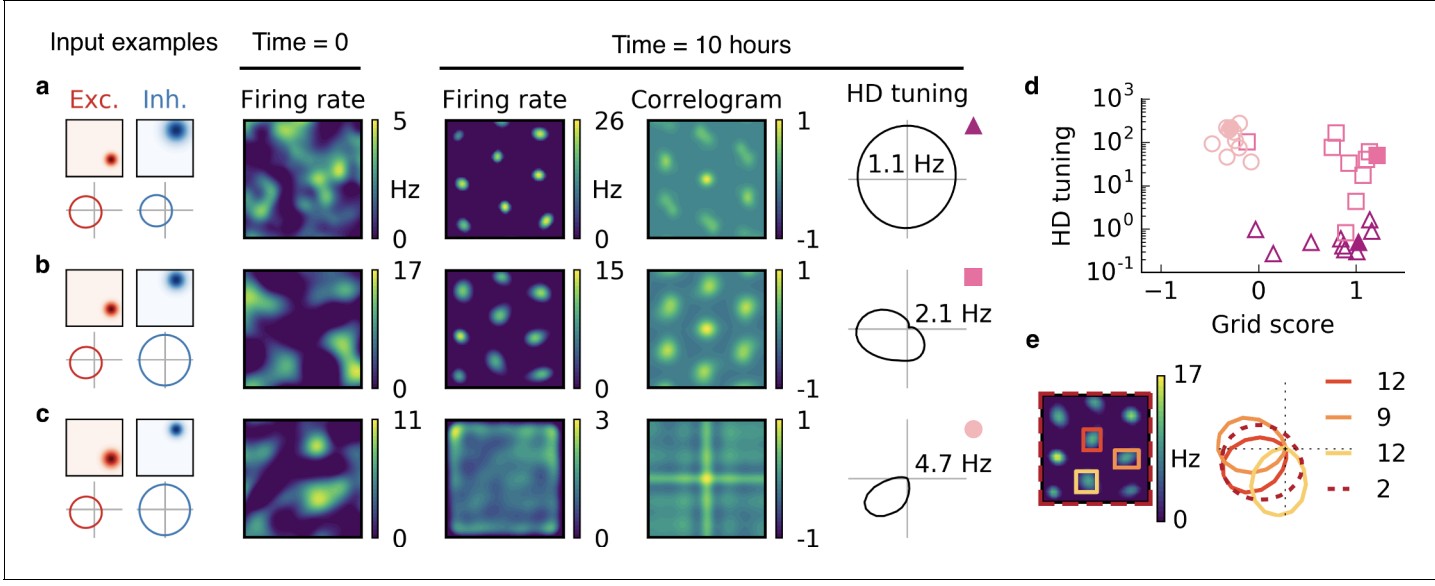

**Figure 6.** Combined spatial and head direction tuning. (a,b,c) Columns from left to right: Spatial tuning and head direction tuning (polar plot) of excitatory and inhibitory input neurons (one example each); spatial firing rate map of the output neuron before learning and after spatial exploration of 10 hr with corresponding autocorrelogram; head direction tuning of the output neuron after learning. The numbers in the polar plots indicate the peak firing rate at the preferred head direction after averaging over space. (a) Wider spatial tuning of inhibitory input neurons than of excitatory input neurons combined with narrower head direction tuning of inhibitory input neurons leads to a grid cell-like firing pattern in space with invariance to head direction, that is, the output neuron fires like a pure grid cell. (b) The same spatial input characteristics combined with head direction-invariant inhibitory input neurons leads to grid cell-like activity in space and a preferred head direction, that is, the output neuron fires like a conjunctive cell. (c) If the spatial tuning of inhibitory input neurons is less smooth than that of excitatory neurons and the concurrent head direction tuning is wider for inhibitory than for excitatory neurons, the output neuron is not tuned to space but to a single head direction, that is, the output neuron fires like a pure head direction cell. (d) Head direction tuning and grid score of 10 simulations of the three cell types. Each symbol represents one realization with random input tuning. The markers correspond to the tuning properties of the input neurons as depicted in (a,b,c): grid cell (triangles), conjunctive cell (squares), head direction cell (circles). The values that correspond to the output cells in (a,b,c) are shown as filled symbols. (e) In our model, the head direction tuning of individual grid fields is sharper than the overall head direction tuning of the conjunctive cell. Depicted is a rate map of a conjunctive cell (left) and the corresponding head direction tuning (right, dashed). For three individual grid fields, indicated with colored squares, the head direction tuning is shown in the same polar plot. The overall tuning of the grid cell (dashed) is a superposition of the tuning of all grid fields. Numbers indicate the peak firing rate (in Hz) averaged individually within each of the four rectangles in the rate map.

DOI: https://doi.org/10.7554/eLife.34560.020

The following figure supplements are available for figure 6:

**Figure supplement 1.** Cells with combined spatial and head direction tuning with input tuning that is given by the sum of 20 randomly located Gaussian ellipsoids.
DOI: https://doi.org/10.7554/eLife.34560.021

**Figure supplement 2.** Head direction tuning of individual grid fields is difficult to assess from grid cells with few firing fields.
DOI: https://doi.org/10.7554/eLife.34560.022

grid fields. Typically, however, these variations remain small enough to preserve an overall head direction tuning of the cell, because individual grid fields tend to align their head direction tuning (compare with *Figure 5—figure supplement 1*, but in three dimensions). Whether or not a narrower head direction of individual grid fields or a different preferred direction for different grid fields is present also in rodents is not resolved (*Figure 6—figure supplement 2*).

## Discussion

We presented a self-organization model that reproduces the experimentally observed spatial and head direction tuning patterns in the hippocampus and adjacent brain regions. Its core mechanism is an interaction of Hebbian plasticity in excitatory synapses and homeostatic Hebbian plasticity in inhibitory synapses (*Vogels et al., 2011*; *D'Amour and Froemke, 2015*). The main prediction of the model is that the spatial autocorrelation structure of excitatory and inhibitory inputs determines – and should thus be predictable from – the output pattern of the cell. Investigations of the tuning of individual cells (*Wertz et al., 2015*) or even synapses (*Wilson et al., 2016*) that project to spatially tuned cells would thus be a litmus test for the proposed mechanism.

### Origin of spatially tuned synaptic input

The origin of synaptic input to spatially tuned cells is not fully resolved (*van Strien et al., 2009*). Given that our model is robust to the precise properties of the input, it is consistent with input from higher sensory areas (*Tanaka, 1996*; *Quiroga et al., 2005*) that could inherit spatial tuning from their sensory tuning in a stable environment (*Arleo and Gerstner, 2000*; *Franzius et al., 2007a*). This is in line with the observation that grid cells lose their firing profiles in darkness (*Chen et al., 2016*; *Pérez-Escobar et al., 2016*) and that the hexagonal pattern rotates when a visual cue card is rotated (*Pérez-Escobar et al., 2016*).

The input could also stem from within the hippocampal formation, where spatial tuning has been observed in both excitatory (*O'Keefe, 1976*) and inhibitory (*Marshall et al., 2002*; *Wilent and Nitz, 2007*; *Hangya et al., 2010*) neurons. For example, the notion that mEC neurons receive input from hippocampal place cells is supported by several studies: Place cells in the hippocampus emerge earlier during development than grid cells in the mEC (*Langston et al., 2010*; *Wills et al., 2010*), grid cells lose their tuning pattern when the hippocampus is deactivated (*Bonnevie et al., 2013*) and both the firing fields of place cells and the spacing and field size of grid cells increase along the dorso-ventral axis (*Jung et al., 1994*; *Brun et al., 2008b*; *Stensola et al., 2012*). Moreover, entorhinal stellate cells, which often exhibit grid-like firing patterns, receive a large fraction of their input from the hippocampal CA2 region (*Rowland et al., 2013*), where many cells are tuned to the location of the animal (*Martig and Mizumori, 2011*).

Inhibition is usually thought to arise from local interneurons – but see (*Melzer et al., 2012*) – suggesting that spatially tuned inhibitory input to mEC neurons originates from the entorhinal cortex itself. Interneurons in mEC display spatial tuning (*Buetfering et al., 2014*; *Savelli et al., 2008*; *Frank et al., 2001*) that could be inherited from hippocampal place cells, other grid cells (*Couey et al., 2013*; *Pastoll et al., 2013*; *Winterer et al., 2017*) or from entorhinal cells with non-grid spatial tuning (*Diehl et al., 2017*; *Hardcastle et al., 2017*). The broader spatial tuning required for the emergence of spatial selectivity could be established, for example by pooling over cells with similar tuning or through a non-linear input-output transformation in the inhibitory circuitry. If inhibitory input is indeed local, the increase in grid spacing along the dorso-ventral axis (*Brun et al., 2008b*) suggests that the tuning of inhibitory interneurons gets smoother along this axis. For smoother tuning functions, fewer neurons are needed to cover the whole environment, in accordance with the decrease in interneuron density along the dorso-ventral axis (*Beed et al., 2013*).

The excitatory input to hippocampal place cells could originate from grid cells in entorhinal cortex (*Figure 5b*), which is supported by anatomical (*van Strien et al., 2009*) and lesion studies (*Brun et al., 2008a*). The required untuned inhibition could arrive from interneurons in the hippocampus proper that often show very weak spatial tuning (*Marshall et al., 2002*). In addition to grid cell input, place cells are also thought to receive inputs from other cell types, such as border cells (*Muessig et al., 2015*) and other brain regions such as the medial septum (*Wang et al., 2015*) .

## Dissociation from continuous attractor network models

The observed spatial tuning patterns have also been explained by other models. In continuous attractor networks (CAN), each cell type could emerge from a specific recurrent connectivity pattern, combined with a mechanism that translates the motion of the animal into shifts of neural activity on an attractor. How the required connectivity patterns – which lie at the core of any CAN model – could emerge is subject to debate (*Widloski and Fiete, 2014*). Our model is qualitatively different in that it does not rely on attractor dynamics in a recurrent neural network, but on experience-dependent plasticity of spatially modulated afferents to an individual output neuron (*Mehta et al., 2000*). A measurable distinction of our model from CAN models is its response to a rapid global reduction of inhibition. While a modification of inhibition typically changes the grid spacing in CAN models of grid cells (*Couey et al., 2013*; *Widloski and Fiete, 2015*), the grid field locations generally remain untouched in our model. The grid fields merely change in size, until inhibition is recovered by inhibitory plasticity (*Figure 7a*). This can be understood by the colocalization of the grid fields and the peaks in the excitatory membrane current (*Figure 7b,c*). A reduction of inhibition leads to an increased protrusion of these excitatory peaks and thus to wider firing fields. Grid patterns in mEC are temporally stable in spite of dopaminergic modulations of GABAergic transmission (*Cilz et al., 2014*) and the spacing of mEC grid cells remains constant during the silencing of inhibitory interneurons (*Miao et al., 2017*). Both observations are in line with our model. Moreover, we found that for localized input tuning, the inhibitory membrane current typically also peaks at the locations of

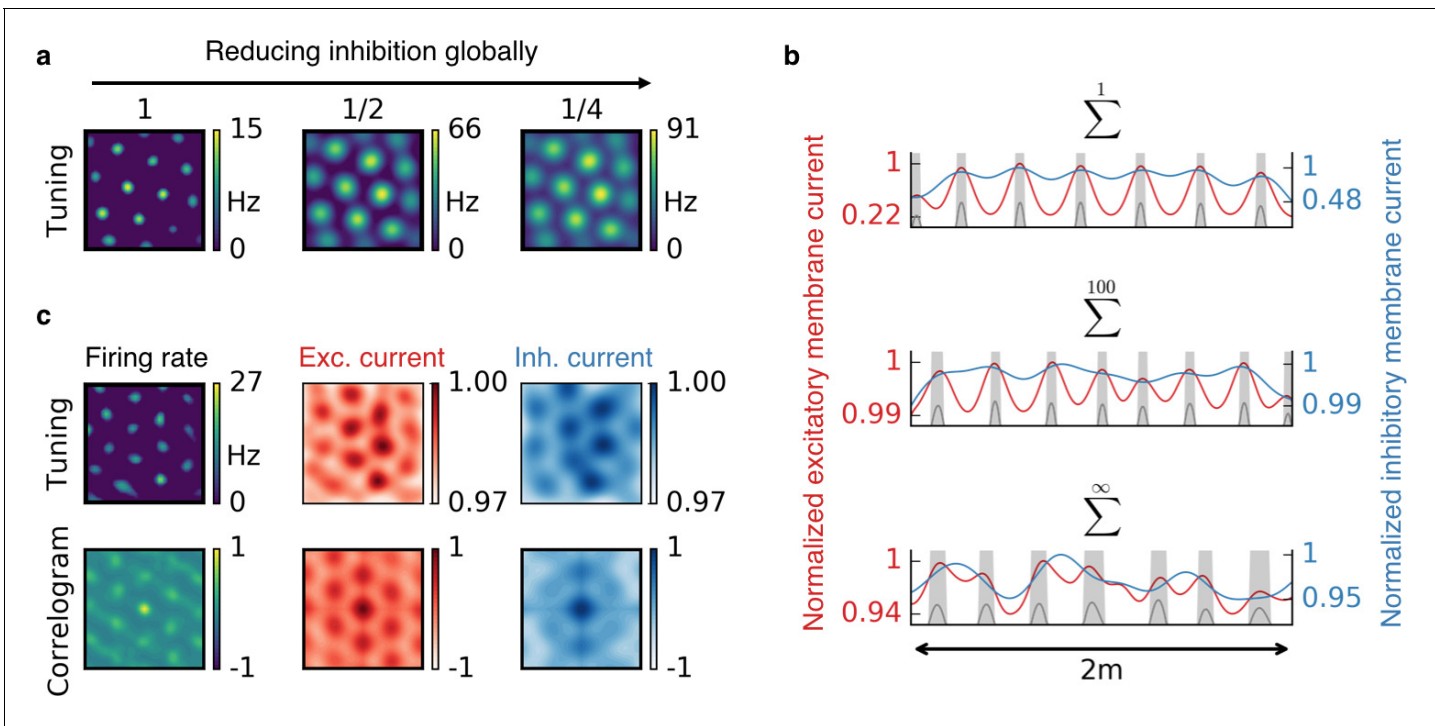

**Figure 7.** Effect of reduced inhibition on grid cell properties. (a) Reducing the strength of inhibitory synapses to a fraction of its initial value (from left to right: 1, 1/2, 1/4) leads to larger grid fields but unchanged grid spacing in our model. In continuous attractor network models, the same reduction of inhibition would affect not only the field size but also the grid spacing. (b) Excitatory (red) and inhibitory (blue) membrane current to a cell with grid-like firing pattern (gray) on a linear track. The currents are normalized to a maximum value of 1. Different rows correspond to different spatial tuning characteristics of the input neurons. From top to bottom: Place cell-like tuning, sparse non-localized tuning (sum of 100 randomly located place fields), dense non-localized tuning (Gaussian random fields). Peaks in excitatory membrane current are co-localized with grid fields (shaded area) for all input statistics. In contrast, the inhibitory membrane current is not necessarily correlated with the grid fields for non-localized input. Moreover, the dynamic range of the membrane currents is reduced for non-localized input. A reduction of inhibition as shown in (a) corresponds to a lowering of the inhibitory membrane current. (c) Excitatory and inhibitory membrane current to a grid cell receiving sparse non-localized input (sum of 100 randomly located place fields) in two dimensions. Top: Tuning of output firing rate, normalized excitatory and inhibitory membrane current. Bottom: Autocorrelograms thereof. The grid pattern is more apparent in the spatial tuning of the excitatory membrane current than in the inhibitory membrane current.
DOI: https://doi.org/10.7554/eLife.34560.023

the grid fields. This co-tuning breaks down for non-localized input (*Figure 7b*). In contrast, CAN models predict that the inhibitory membrane current has the same periodicity as the grid (*Schmidt-Hieber and Häusser, 2013*), but possibly phase shifted.

The grid patterns of topologically nearby grid cells in the mEC typically have the same orientation and spacing but different phases (*Hafting et al., 2005*). Moreover, the coupling between anatomically nearby grid cells – for example their difference in spatial phase – is more stable to changes of the environment than the firing pattern of individual grid cells (*Yoon et al., 2013*). These properties are immanent to CAN models. In contrast, single cell models (*Burgess et al., 2007*; *Kropff and Treves, 2008*; *Castro and Aguiar, 2014*; *Stepanyuk, 2015*; *Dordek et al., 2016*; *D'Albis and Kempter, 2017*; *Monsalve-Mercado and Leibold, 2017*) require additional mechanisms to develop a coordination of neighboring grid cells. The challenge for any mechanism is to correlate the grid orientations, but leave the grid phases uncorrelated. The most obvious candidate, recurrent connections among different grid cells (*Si et al., 2012*), requires an intricate combination of mechanisms to perform this balancing act. We assume that an appropriate recurrent connectivity would not be simpler in our model.

CAN models predict that all grid fields in a conjunctive (grid x head direction) cell have the same head direction tuning, whereas our model predicts that there could be differences between different grid fields (*Figure 6e*). Our preliminary analysis suggests that an in-depth evaluation would require data for central grid fields without trajectory biases (*Figure 6—figure supplement 2*), which are at present not publicly available.

In addition, CAN models require that conjunctive (grid x head direction) cells are positively modulated by running speed. Such modulation has been observed in experiments (*Kropff et al., 2015*). In our model, we could introduce a running speed dependence, for example as a global modulation of the input signals. We expect that in this case, the output neuron would inherit speed tuning from the input but would otherwise develop similar spatial tuning patterns.

A recent analysis has shown that periodic firing of entorhinal cells in rats that move on a linear track can be assessed as slices through a hexagonal grid (*Yoon et al., 2016*), which arises naturally in a two-dimensional CAN model. In our model, we would obtain slices through a hexagonal grid if the rat learns the output pattern in two dimensions and afterwards is constrained to move on a linear track that is part of the same arena. If the rat learns the firing pattern on the linear track from scratch, the firing fields would be periodic.

## Rapid appearance and rearrangement of grids

Models that learn grid cells from spatially tuned input do not have to assume a preexisting connectivity pattern or specific mechanisms for path integration (*Burgess et al., 2007*), but are challenged by the fast emergence of hexagonal firing patterns in unfamiliar environments (*Hafting et al., 2005*). Most plasticity-based models require slow learning, such that the animal explores the whole arena before significant synaptic changes occur. Therefore, grid patterns typically emerge slower than experimentally observed (*Dordek et al., 2016*). This delay is particularly pronounced in models that require an extensive exploration of both space and movement direction (*Kropff and Treves, 2008*; *Franzius et al., 2007a*; *D'Albis and Kempter, 2017*). In contrast to these models, which give center stage to the temporal statistics of the animal's movement, our approach relies purely on the spatial statistics of the input and is hence insensitive to running speed.

For the mechanism we suggested, the self-organization was very robust and allowed rapid pattern formation on short time scales, similar to those observed in rodents (*Figure 3*). This speed could be further increased by accelerated reactivation of previous experiences during periods of rest (*Lee and Wilson, 2002*). By this means, the exploration time and the time it takes to activate all input patterns could be decoupled, leading to a much faster emergence of grid cells in all trajectory-independent models with associative learning. Other models that explain the emergence of grid patterns from place cell input through synaptic depression and potentiation also develop grid cells in realistic times (*Castro and Aguiar, 2014*; *Stepanyuk, 2015*; *Monsalve-Mercado and Leibold, 2017*). These models differ from ours in that they do not require inhibition, but instead specific forms of rate-dependent synaptic depression and potentiation that change the synaptic weights such that place cell-like input leads to grid cell-like output. How these models generalize to potentially non-localized input is yet to be shown.

Learning the required connectivity in CAN models can take a long time (*Widloski and Fiete, 2014*). However, as soon as the required connectivity and translation mechanism is established, a grid pattern would be observed immediately, even in a novel room. For different rooms this pattern could have different phases and orientations, but similar grid spacing (*Fyhn et al., 2007*). Similarly, we found that room switches in our model lead to grid patterns of the same grid spacing but different phases and orientations. The pattern emerges rapidly, but is not instantaneously present (*Figure 3—figure supplement 2*). It would be interesting to study whether rotation of a fraction of the input would lead to a bimodal distribution of grid rotations: No rotation and co-rotation with the rotated input, as recently observed in experiments where distal cues were rotated but proximal cues stayed fixed (*Savelli et al., 2017*).

Recently, it was discovered that in an arena separated by a wall, single grid cells form two independent grid patterns – one on each side – that coalesce once the wall is removed (*Wernle et al., 2018*; Rosay et al., in preparation). This coalescence is local, that is, grid fields close to the partition wall readjust, whereas grid fields far away do not change their locations. Feedforward models like ours can explain such a local rearrangement (*Figure 4*; Rosay et al., in preparation).

### Boundary effects

Experiments show that the pattern and the orientation of grid cells is influenced by the geometry of the environment. In a quadratic arena, the orientation of grid cells tends to align – with a small offset – to one of the box axes (*Stensola et al., 2015*). In trapezoidal arenas, the hexagonality of grids is distorted (*Krupic et al., 2015a*). We considered quadratic and circular arenas with rat trajectories from behavioral experiments and found that the boundaries also distort the grid pattern in our simulations, particularly for localized inputs (*Figure 2—figure supplement 3*). In trapezoidal geometries, we expect this to lead to non-hexagonal grids. However, we did not observe a pronounced alignment to quadratic boundaries if the input place fields were randomly located (*Figure 2—figure supplement 3*).

### Conclusion

We found that interacting excitatory and inhibitory plasticity serves as a simple and robust mechanism for rapid self-organization of stable and symmetric patterns from spatially modulated feedforward input. The suggested mechanism ports the robust pattern formation of attractor models from the neural to the spatial domain and increases the speed of self-organization of plasticity-based mechanisms to time scales on which the spatial tuning of neurons is typically measured. It will be interesting to explore how recurrent connections between output cells can help to understand the role of local inhibitory (*Couey et al., 2013*; *Pastoll et al., 2013*) and excitatory connections (*Winterer et al., 2017*) and the presence or absence of topographic arrangements of spatially tuned cells (*O'Keefe et al., 1998*; *Stensola et al., 2012*; *Giocomo et al., 2014*). We illustrated the properties and requirements of the model in the realm of spatial representations. As invariance and selectivity are ubiquitous properties of receptive fields in the brain, the interaction of excitatory and inhibitory synaptic plasticity could also be essential to form stable representations from sensory input in other brain areas (*Constantinescu et al., 2016*; *Clopath et al., 2016*).

# Materials and methods

## Code availability

The code for reproducing the essential findings of this article is available at https://github.com/sim-web/spatial_patterns (*Weber, 2018*) under the GNU General Public License v3.0. A copy is archived at https://github.com/elifesciences-publications/spatial_patterns.

## Network architecture and neuron model

We study a feedforward network where a single output neuron receives synaptic input from $N_{\mathrm{E}}$ excitatory and $N_{\mathrm{I}}$ inhibitory neurons (*Figure 1a*) with synaptic weight vectors $\mathbf{w}^{\mathrm{E}} \in \mathbb{R}^{N_{\mathrm{E}}}$, $\mathbf{w}^{\mathrm{I}} \in \mathbb{R}^{N_{\mathrm{I}}}$ and spatially tuned input rates $\mathbf{r}^{\mathrm{E}}(\mathbf{x}) \in \mathbb{R}^{\mathbf{N}_{\mathrm{E}}}$, $\mathbf{r}^{\mathrm{I}}(\mathbf{x}) \in \mathbb{R}^{\mathbf{N}_{\mathrm{I}}}$, respectively. Here $\mathbf{x} \in \mathbb{R}^{\mathrm{dimensions}}$ denotes the location and later also the head direction of the animal. For simplicity and to allow a mathematical

analysis we use a rate-based description for all neurons. The firing rate of the output neuron is given by the rectified sum of weighted excitatory and inhibitory inputs:

$$r^{\text{out}}(\mathbf{x}(t)) = \left[ \sum_{i=1}^{N_{\text{E}}} w_i^{\text{E}}(t) r_i^{\text{E}}(\mathbf{x}(t)) - \sum_{j=1}^{N_{\text{I}}} w_j^{\text{I}}(t) r_j^{\text{I}}(\mathbf{x}(t)) \right]_+ , \tag{1}$$

where $[\cdot]_+$ denotes a rectification that sets negative firing rates to zero. To comply with the notion of excitation and inhibition, all weights are constrained to be positive. In most simulations we use $N_{\text{E}} = 4N_{\text{I}}$. Simulation parameters are shown in *Tables 1–3* for the main figures and in *Tables 4–6* for the supplementary figures.

## Excitatory and inhibitory plasticity

In each unit time step ($\Delta t = 1$), the excitatory weights are updated according to a Hebbian rule:

$$\Delta \mathbf{w}^{\text{E}} = \eta_{\text{E}} \mathbf{r}^{\text{E}}(\mathbf{x}) r^{\text{out}}(\mathbf{x}) \quad \text{(and normalization)}. \tag{2}$$

The excitatory learning rate $\eta_{\text{E}}$ is a constant that we chose individually for each simulation. To avoid unbounded weight growth, we use a quadratic multiplicative normalization, that is, we keep the sum of the squared weights of the excitatory population $\sum_{i=1}^{N_{\text{E}}} (w_i^{\text{E}})^2$ constant at its initial value, by rescaling the weights after each unit time step. However, synaptic weight normalization is not a

**Table 1.** Parameters for excitatory inputs for all figures in the manuscript.
$N_{\text{E}}^{\text{f}} = \infty$ indicates that the excitatory input is a Gaussian random field.

| | $[\sigma_{\text{E},x}, \sigma_{\text{E},y}, \sigma_{\text{E},z}]$ | $N_{\text{E}}$ | $\eta_{\text{E}}$ | $w^{\text{E,init}}$ | $N_{\text{E}}^{\text{f}}$ |
|---|---|---|---|---|---|
| *Figure 1b* | 0.05 | 2000 | $2 \times 10^{-6}$ | 1 | $\infty$ |
| *Figure 1c* | 0.08 | 2000 | $2 \times 10^{-6}$ | 1 | $\infty$ |
| *Figure 1d* | 0.06 | 2000 | $2 \times 10^{-6}$ | 1 | $\infty$ |
| *Figure 1f* | 0.04 | 160 | $2 \times 10^{-6}$ | 1 | 1 |
| *Figure 1g* | 0.03 | 1600 | $3.6 \times 10^{-5}$ | 1 | 1 |
| *Figure 1h* | 0.03 | 10000 | $3.5 \times 10^{-7}$ | 1 | $\infty$ |
| *Figure 2a* | [0.05, 0.05] | 4900 | $6.7 \times 10^{-5}$ | 1 | 1 |
| *Figure 2b* | [0.05, 0.05] | 4900 | $2 \times 10^{-6}$ | 1 | 100 |
| *Figure 2c* | [0.05, 0.05] | 4900 | $6 \times 10^{-6}$ | 1 | $\infty$ |
| *Figure 3a–d* | [0.05, 0.05] | 4900 | $2 \times 10^{-4}$ | 1 | 1 |
| *Figure 4* | [0.05, 0.05] | $2 \times 4900$ | $1.3 \times 10^{-4}$ | 1 | 1 |
| *Figure 5a* | [0.07, 0.07] | 4900 | $6 \times 10^{-6}$ | 0.5 | $\infty$ |
| *Figure 5b* | [0.07, 0.07] | 400 | $1.3 \times 10^{-4}$ | 1 | 1 |
| *Figure 5c* | [0.05, 0.05] | 4900 | $1.1 \times 10^{-6}$ | 0.0455 | $\infty$ |
| *Figure 5d* | [0.08, 0.08] | 4900 | $6 \times 10^{-6}$ | 0.5 | $\infty$ |
| *Figure 5e* | [0.05, 0.05] | 4900 | $6.7 \times 10^{-5}$ | 1 | 1 |
| *Figure 6a* | [0.07, 0.07, 0.2] | 37500 | $1.5 \times 10^{-5}$ | 1 | 1 |
| *Figure 6b* | [0.08, 0.08, 0.2] | 50000 | $10^{-5}$ | 1 | 1 |
| *Figure 6c* | [0.1, 0.1, 0.2] | 50000 | $10^{-5}$ | 1 | 1 |
| *Figure 7a* | [0.05, 0.05] | 4900 | $6.7 \times 10^{-5}$ | 1 | 1 |
| *Figure 7b* | 0.04 | 2000 | $5 \times 10^{-5}$ | 1 | 1 |
| | 0.04 | 2000 | $5 \times 10^{-7}$ | 1.0 | 100 |
| *Figure 7c* | 0.05 | 2000 | $5 \times 10^{-6}$ | 0.5 | $\infty$ |
| | [0.05, 0.05] | 4900 | $2 \times 10^{-6}$ | 1 | 100 |
| *Figure 8b* | 0.03 | 800 | $3.3 \times 10^{-5}$ | 1 | 1 |

DOI: https://doi.org/10.7554/eLife.34560.024

**Table 2.** Parameters for inhibitory inputs for all figures in the manuscript.
indicates that the inhibitory input is a Gaussian random field. We denote spatially untuned inhibition with: $\sigma_I = \infty$.

| | $[\sigma_{I,x}, \sigma_{I,y}, \sigma_{I,z}]$ | $N_I$ | $\eta_I$ | $w^{I,init}$ | $N_I^f$ |
|---|---|---|---|---|---|
| *Figure 1b* | 0.12 | 500 | $2 \times 10^{-5}$ | 4:4 | $\infty$ |
| *Figure 1c* | 0.07 | 2000 | $2 \times 10^{-5}$ | 1.1 | $\infty$ |
| *Figure 1d* | $\infty$ | 500 | $2 \times 10^{-5}$ | 4.39 | $\infty$ |
| *Figure 1f* | 0.13 | 40 | $2 \times 10^{-5}$ | 1.31 | 1 |
| *Figure 1g* | From 0.08 to 0.3 in 0.02 steps | 400 | $3.6 \times 10^{-4}$ | **Equation 111** | 1 |
| *Figure 1h* | From 0.08 to 0.3 in 0.02 steps | 2500 | $7 \times 10^{-6}$ | 4.03 | $\infty$ |
| *Figure 2a* | [0.1, 0.1] | 1225 | $2.7 \times 10^{-4}$ | 1.5 | 1 |
| *Figure 2b* | [0.1, 0.1] | 1225 | $8 \times 10^{-6}$ | 1.52 | 100 |
| *Figure 2c* | [0.1, 0.1] | 1225 | $6 \times 10^{-5}$ | 4.0 | $\infty$ |
| *Figure 3a–d* | [0.1, 0.1] | 1225 | $8 \times 10^{-4}$ | 1.5 | 1 |
| *Figure 4* | [0.1, 0.1] | $2 \times 1225$ | $5.3 \times 10^{-4}$ | 1.51 | 1 |
| *Figure 5a* | [$\infty$, $\infty$] | 1225 | $6 \times 10^{-5}$ | 2 | $\infty$ |
| *Figure 5b* | [$\infty$, $\infty$] | 1 | $5.3 \times 10^{-4}$ | 69.5 | 1 |
| *Figure 5c* | [0.049, 0.049] | 1225 | $4.4 \times 10^{-5}$ | 0.175 | $\infty$ |
| *Figure 5d* | [0.3, 0.07] | 1225 | $6 \times 10^{-5}$ | 2 | $\infty$ |
| *Figure 5e* | [0.049, 0.049] | 4900 | $2.7 \times 10^{-4}$ | 1.02 | 1 |
| | [0.2, 0.1]; [0.1, 0.2] | 1225 | $2.7 \times 10^{-4}$ | 1.04 | 1 |
| | [2, 0.1]; [0.1, 2] | 1225 | $2.7 \times 10^{-4}$ | 2.74 | 1 |
| | [2, 0.2]; [0.2, 2] | 1225 | $2.7 \times 10^{-4}$ | 1.38 | 1 |
| | [0.1, 0.1] | 1225 | $2.7 \times 10^{-4}$ | 1.5 | 1 |
| | [0.2, 0.2] | 1225 | $2.7 \times 10^{-4}$ | 0.709 | 1 |
| | [2, 2] | 1225 | $2.7 \times 10^{*-4}$ | 0.259 | 1 |
| | [0.1, 0.049]; [0.049, 0.1] | 1225 | $2.7 \times 10^{-4}$ | 2.48 | 1 |
| | [0.2, 0.049]; [0.049, 0.2] | 1225 | $2.7 \times 10^{-4}$ | 1.74 | 1 |
| *Figure 6a* | [2, 0.049]; [0.049, 2] | 1225 | $2.7 \times 10^{-4}$ | 5.56 | 1 |
| | [0.15, 0.15, 0.2] | 9375 | $1.5 \times 10^{-4}$ | 1.55 | 1 |
| *Figure 6b* | [0.12, 0.12, 1.5] | 3125 | $10^{-4}$ | 5.68 | 1 |
| *Figure 6c* | [0.09, 0.09, 1.5] | 12500 | $10^{-4}$ | 2.71 | 1 |
| *Figure 6d* | Same as | **Figure 6a,b,c** | | | |
| *Figure 7a* | [0.1, 0.1] | 1225 | $2.7 \times 10^{-4}$ | 1.5 | 1 |
| *Figure 7b* | 0.12 | 500 | $5 \times 10^{-4}$ | 1.6 | 1 |
| | 0.12 | 500 | $5 \times 10^{-6}$ | 1.62 | 100 |
| *Figure 7c* | 0.12 | 500 | $5 \times 10^{-5}$ | 1.99 | $\infty$ |
| | [0.1, 0.1] | 1225 | $8 \times 10^{-6}$ | 1.52 | 100 |
| *Figure 8b* | 0.1 | varied | varied | varied | 1 |

DOI: https://doi.org/10.7554/eLife.34560.025

necessary ingredient for the emergence of firing patterns (*Figure 2—figure supplement 4*). We model inhibitory synaptic plasticity using a previously suggested learning rule (*Vogels et al., 2011*):

$$\Delta \mathbf{w}^I = \eta_I \mathbf{r}^I(\mathbf{x})(r^{out}(\mathbf{x}) - \rho_0), \tag{3}$$

with inhibitory learning rate $\eta_I$ and target rate $\rho_0$ = 1 Hz. Negative inhibitory weights are set to zero.

**Table 3.** Simulation time $t_{\mathrm{sim}}$ and system size $L$ for all figures in the manuscript.

|  | $t_{\mathrm{sim}}$ | $L$ |
| --- | --- | --- |
| *Figure 1b* | 2,000,000 | 2 |
| *Figure 1c* | 2,000,000 | 2 |
| *Figure 1d* | 400,000 | 2 |
| *Figure 1f* | 20,000,000 | 2 |
| *Figure 1g* | 80,000,000 | 14 |
| *Figure 1h* | 40,000,000 | 10 |
| *Figure 2a,b,c* | 1,800,000 | 1 |
| *Figure 3a,b,c,d* | 540,000 | 1 |
| *Figure 4* | 1,800,000 | 1 |
| *Figure 5a,c,d,e* | 1,800,000 | 1 |
| *Figure 5b* | 180,000 | 1 |
| *Figure 6a,b,c,d* | 1,800,000 | 1 |
| *Figure 7a,c* | 1,800,000 | 1 |
| *Figure 7b* | 400,000 | 2 |
| *Figure 8b* | 40,000,000 | 3 |

DOI: https://doi.org/10.7554/eLife.34560.026

## Rat trajectory

In the linear track model (one dimension, *Figures 1* and *7*), we create artificial run-and-tumble trajectories $x(t)$ constrained on a line of length $L$ with constant velocity $v$ = 1 cm per unit time step and persistence length $L/2$ (Appendix 1).

In the open arena model (two dimensions, *Figures 2*, *3*, *5* and *7*), we use trajectories $\mathbf{x}(t)$ from behavioral data (*Sargolini et al., 2006b*) of a rat that moved in a 1 m × 1 m quadratic enclosure (Appendix 1). In the simulations with a separation wall (*Figure 4*), we create trajectories as a two-dimensional persistent random walk (Appendix 1). In the model for neurons with head direction tuning (three dimensions, *Figure 6*), we use the same behavioral trajectories as in two dimensions and model the head direction as noisily aligned to the direction of motion (Appendix 1).

**Table 4.** Parameters for excitatory inputs in supplement figures.
$N_{\mathrm{E}}^{\mathrm{f}} = \infty$ indicates that the excitatory input is a Gaussian random field.

|  | $[\sigma_{\mathrm{E},x}, \sigma_{\mathrm{E},y}, \sigma_{\mathrm{E},z}]$ | $N_{\mathrm{E}}$ | $\eta_{\mathrm{E}}$ | $w^{\mathrm{E,init}}$ | $N_{\mathrm{E}}^{\mathrm{f}}$ |
| --- | --- | --- | --- | --- | --- |
| *Figure 1—figure supplement 1* | 0.04 | 2000 | $5 \times 10^{-7}$ | 1 | varied |
| *Figure 1—figure supplement 2* | see | caption |  |  |  |
| *Figure 2—figure supplement 1* | [0.05, 0.05] | 4900 | $6.7 \times 10^{-5}$ | 1 | 1 |
| *Figure 2—figure supplement 3* | [0.05, 0.05] | 4900 | $6.7 \times 10^{-5}$ | 1 | 1 |
| *Figure 2—figure supplement 4* | [0.05, 0.05] | 4900 | $2 \times 10^{-4}$ | 1 | 1 |
| *Figure 2—figure supplement 6* | see | caption |  |  |  |
| *Figure 2—figure supplement 2* | [0.05, 0.05] | 4900 | $3.3 \times 10^{-5}$ | 1 | 1 |
| *Figure 3—figure supplement 1* | see | caption |  |  |  |
| *Figure 3—figure supplement 3* | see | caption |  |  |  |
| *Figure 3—figure supplement 2* | [0.05, 0.05] | 4900 | $1.3 \times 10^{-4}$ | 1 | 1 |
| *Figure 6—figure supplement 1* | see | caption |  |  |  |

DOI: https://doi.org/10.7554/eLife.34560.027

**Table 5.** Parameters for inhibitory inputs in supplement figures.

$N_{\mathrm{I}}^{\mathrm{f}} = \infty$ indicates that the inhibitory input is a Gaussian random field. We denote spatially untuned inhibition with: $\sigma_{\mathrm{I}} = \infty$.

|  | $[\sigma_{\mathrm{I},x}, \sigma_{\mathrm{I},y}, \sigma_{\mathrm{I},z}]$ | $N_{\mathrm{I}}$ | $\eta_{\mathrm{I}}$ | $w^{\mathrm{I,init}}$ | $N_{\mathrm{I}}^{\mathrm{f}}$ |
|---|---|---|---|---|---|
| *Figure 1—figure supplement 1* | 0.12 | 500 | $5 \times 10^{-6}$ | 1.61 | varied |
| *Figure 1—figure supplement 2* | see | caption | | | |
| *Figure 2—figure supplement 1* | [0.1, 0.1] | 1225 | $2.7 \times 10^{-4}$ | 1.5 | 1 |
| *Figure 2—figure supplement 3* | [0.1, 0.1] | 1225 | $2.7 \times 10^{-4}$ | 1.5 | 1 |
| *Figure 2—figure supplement 4* | [0.1, 0.1] | 1225 | $8 \times 10^{-4}$ | 1.5 | 1 |
| *Figure 2—figure supplement 6* | see | caption | | | |
| *Figure 2—figure supplement 2* | [0.1, 0.1] | 1225 | $5.3 \times 10^{-6}$ | 0.03 | 50 |
| *Figure 3—figure supplement 1* | see | caption | | | |
| *Figure 3—figure supplement 3* | see | caption | | | |
| *Figure 3—figure supplement 2* | [0.1, 0.1] | 1225 | $5.3 \times 10^{-4}$ | 1.5 | 1 |
| *Figure 6—figure supplement 1* | see | caption | | | |

DOI: https://doi.org/10.7554/eLife.34560.028

## Spatially tuned inputs

The firing rates of excitatory and inhibitory synaptic inputs $r_i^{\mathrm{E}}, r_j^{\mathrm{I}}$ are tuned to the location $\mathbf{x}$ of the animal. In the following, we use $x$ and $y$ for the first and second spatial dimension and $z$ for the head direction.

For place field-like input, we use Gaussian tuning functions with standard deviation $\sigma_{\mathrm{E}}$, $\sigma_{\mathrm{I}}$ for the excitatory and inhibitory population, respectively. In *Figure 5* the standard deviation is chosen independently along the $x$ and $y$ direction. The centers of the Gaussians are drawn randomly from a distorted lattice (*Figure 2—figure supplement 5*). This way we ensure random but spatially dense tuning. The lattice contains locations outside the box to reduce boundary effects.

For sparse non-localized input with $N_{\mathrm{P}}^{\mathrm{f}}$ fields per neuron of population P, we first create $N_{\mathrm{P}}^{\mathrm{f}}$ distorted lattices, each with $N_{\mathrm{P}}$ locations. We then assign $N_{\mathrm{P}}^{\mathrm{f}}$ of the resulting $N_{\mathrm{P}}^{\mathrm{f}} N_{\mathrm{P}}$ locations at random and without replacement to each input neuron (see also Appendix 1).

For dense non-localized input, we convolve Gaussians with white noise and increase the resulting signal to noise ratio by setting the minimum to zero and the mean to 0.5 (Appendix 1). The Gaussian convolution kernels have different standard deviations for different populations. For each input neuron we use a different realization of white noise. This results in arbitrary tuning functions of the same

**Table 6.** Simulation time $t_{\mathrm{sim}}$ and system size $L$ for supplement figures.

|  | $t_{\mathrm{sim}}$ | $L$ |
|---|---|---|
| *Figure 1—figure supplement 1* | 48,000,000 | 1 |
| *Figure 1—figure supplement 2* | see | caption |
| *Figure 2—figure supplement 1* | 1,800,000 | 0.5 |
| *Figure 2—figure supplement 3* | 1,800,000 | 0.5 |
| *Figure 2—figure supplement 4* | 180,000 | 0.5 |
| *Figure 2—figure supplement 6* | see | caption |
| *Figure 2—figure supplement 2* | 1,800,000 | 0.5 |
| *Figure 3—figure supplement 1* | see | caption |
| *Figure 3—figure supplement 3* | see | caption |
| *Figure 3—figure supplement 2* | 1,800,000 | 0.5 |
| *Figure 6—figure supplement 1* | see | caption |

DOI: https://doi.org/10.7554/eLife.34560.029

autocorrelation length as the – potentially asymmetric – Gaussian convolution kernel. For grid cell-like input, we place Gaussians of standard deviation $\sigma_E$ on the nodes of perfect hexagonal grids whose spacing and orientation is variable. In *Figure 5*b we draw the grid spacing of each input from a normal distribution of mean $6\sigma_E$ and standard deviation $\sigma_E/6$. The grid orientation was drawn from a uniform distribution between $-30$ and $30$ degrees.

For input with combined spatial and head direction tuning, we use the Gaussian tuning curves described above for the spatial tuning and von Mises distributions along the head direction dimension (Appendix 1).

For all input tunings, the standard deviation of the firing rate is of the same order of magnitude as the mean firing rate (Appendix 1).

## Initial synaptic weights and global reduction of inhibition

We specify a mean for the initial excitatory and inhibitory weights, respectively, and randomly draw each synaptic weight from the corresponding mean $\pm 5\%$. The excitatory mean is chosen such that the output neuron would fire above the target rate everywhere in the absence of inhibition; we typically take this mean to be 1 (*Table 1* and Appendix 1). The mean inhibitory weight is then determined such that the output neuron would fire close to the target rate, if all the weights were at their mean value (*Table 2* and Appendix 1). Choosing the weights this way ensures that initial firing rates are random, but neither zero everywhere, nor inappropriately high. We model a global reduction of inhibition by scaling all inhibitory weights by a constant factor, after the grid has been learned.

## Mathematical analysis of the learning rules

In the following, we derive the spacing of periodic firing patterns as a function of the simulation parameters for the linear track.

We first show that homogeneous weights, chosen such that the output neuron fires at the target rate, are a fixed point for the time evolution of excitatory and inhibitory weights under the assumption of slow learning. We then perturb this fixed point and study the time evolution of the perturbations in Fourier space. The translational invariance of the input overlap leads to decoupling of spatial frequencies and leaves a two-dimensional dynamical system for each spatial frequency. For smoother spatial tuning of inhibitory input than excitatory input, the eigenvalue spectrum of the dynamical system has a unique maximum, which indicates the most unstable spatial frequency. This frequency accurately predicts the grid spacing. We first consider place cell-like input (Gaussians) and then non-localized input (Gaussians convolved with white noise).

At the end of the analysis, you will find a glossary of the notation. Whenever we use P as a sub- or superscript instead of E or I, this implies that the equation holds for neurons of the excitatory and the inhibitory population.

The analysis is written as a detailed and comprehensible walk-through. The reader who is interested only in the result can jump to *Equations 78 and 104*.

### Assumption of slow learning

The firing rate of the output neuron is the weighted sum of excitatory and inhibitory input rates:

$$r^{\text{out}} = \left[ \mathbf{w}^E \cdot \mathbf{r}^E - \mathbf{w}^I \cdot \mathbf{r}^I \right]_+, \tag{4}$$

where $[\ldots]_+$ indicates that negative firing rates are set to zero.

Written as a differential equation, the excitatory learning rule with quadratic multiplicative normalization is given by:

$$\frac{d\mathbf{w}^E}{dt} = \eta_E \left( \mathbb{1} - \frac{\mathbf{w}^E \mathbf{w}^{E\mathsf{T}}}{\|\mathbf{w}^E\|^2} \right) \mathbf{r}^E r^{\text{out}}, \tag{5}$$

where $\mathbb{1}$ is the $N_E \times N_E$ identity matrix. The projection operator $\frac{\mathbf{w}^E \mathbf{w}^{E\mathsf{T}}}{\|\mathbf{w}^E\|^2}$ ensures that the weights are constrained to remain on the hypersphere whose radius is determined by the initial value of the sum of the squares of all excitatory weights (*Miller and MacKay, 1994*). The inhibitory learning rule is given by:

$$\frac{d\mathbf{w}^{\mathrm{I}}}{dt} = \eta_{\mathrm{I}}\mathbf{r}^{\mathrm{I}}\left(r^{\mathrm{out}} - \rho_0\right). \tag{6}$$

We assume that the rat will learn slowly, such that it forages through the environment before significant learning (i.e. weight change) occurs. Therefore we can coarsen the time scale and rewrite *Equation 5 and 6* as

$$\frac{d\mathbf{w}^{\mathrm{E}}}{dt} = \eta_{\mathrm{E}}\left\langle\left(\mathbb{1} - \frac{\mathbf{w}^{\mathrm{E}}\mathbf{w}^{\mathrm{E\,T}}}{\|\mathbf{w}^{\mathrm{E}}\|^2}\right)\mathbf{r}^{\mathrm{E}}r^{\mathrm{out}}\right\rangle_x \tag{7}$$

and

$$\frac{d\mathbf{w}^{\mathrm{I}}}{dt} = \eta_{\mathrm{I}}\left\langle\mathbf{r}^{\mathrm{I}}\left(r^{\mathrm{out}} - \rho_0\right)\right\rangle_x, \tag{8}$$

respectively, where the spatial average, $\langle\ldots\rangle_x$, is defined as

$$\langle(\ldots)\rangle_x = \frac{1}{L}\int_{-L/2}^{+L/2}(\ldots)\,dx \tag{9}$$

and $L$ is the length of the linear track.

## High density assumption and continuum limit for place cell-like input

We assume a high density of input neurons and formulate the system in continuous variables. More precisely, we assume the distance between two neighboring firing fields to be much smaller than the width of the firing fields, that is, $L/N_P \ll \sigma_{\mathrm{P}}$. Furthermore, we assume that the linear track is very long compared with the width of the firing fields, that is, $\sigma_{\mathrm{P}} \ll L$.

We replace the neuron index with the continuous variable $\mu$ and denote the weight $w_\mu^{\mathrm{P}}$ and the tuning function $r^{\mathrm{P}}(\mu,x)$ associated with a place field that is centered at $\mu$ in the continuum limit as:

$$w_i^{\mathrm{P}} \to w^{\mathrm{P}}(\mu) \text{ and } r_i^{\mathrm{P}}(x) \to r^{\mathrm{P}}(\mu,x). \tag{10}$$

The distance between two neighboring place fields is given by $\Delta\mu = L/N_{\mathrm{P}}$. Thus, for sums over all neurons we get the following integral in the continuum limit:

$$\sum_{i=1}^{N_{\mathrm{P}}} f_i = \frac{1}{\Delta\mu}\sum_{i=1}^{N_{\mathrm{P}}} f_i\,\Delta\mu \to \frac{N_{\mathrm{P}}}{L}\int_{-L/2}^{+L/2} f(\mu)\,d\mu. \tag{11}$$

We will switch between the discrete and continuous formulations, using whatever is more convenient.

For place cell-like input we take Gaussian tuning curves:

$$r_i^{\mathrm{P}}(x) = \alpha_{\mathrm{P}}\exp\left\{-\frac{(x-\mu_i)^2}{2\sigma_{\mathrm{P}}^2}\right\}, \tag{12}$$

with height $\alpha_{\mathrm{P}}$ and standard deviation $\sigma_{\mathrm{P}}$. Thus, in the continuum limit we get:

$$r_i^{\mathrm{P}}(x) \to r^{\mathrm{P}}(\mu,x) = r^{\mathrm{P}}(|x-\mu|) = \alpha_{\mathrm{P}}\exp\left\{-\frac{(x-\mu)^2}{2\sigma_{\mathrm{P}}^2}\right\}. \tag{13}$$

Because of the translational invariance of $r^{\mathrm{P}}(\mu,x)$, integration over space gives the same result as integration over all center locations and the mean of all inputs is the same:

$$\left\langle r_i^{\mathrm{P}}(x)\right\rangle_x = \left\langle r^{\mathrm{P}}(\mu,x)\right\rangle_x \tag{14}$$

$$= \frac{1}{L}\int_{-L/2}^{+L/2} r^{\mathrm{P}}(\mu,x)\,dx \tag{15}$$

$$= \frac{1}{L} \int_{-L/2}^{+L/2} r^{\mathrm{P}}(\mu,x)\,\mathrm{d}\mu \approx \frac{\alpha_{\mathrm{P}}}{L}\sqrt{2\pi\sigma_{\mathrm{P}}^2} = M_{\mathrm{P}}/L \tag{16}$$

where we introduced $M_{\mathrm{P}} := \alpha_{\mathrm{P}}\sqrt{2\pi\sigma_{\mathrm{P}}^2}$ for the area under the tuning curves. Accordingly, we get a summarized input activity that is independent of location:

$$\sum_{i=1}^{N_{\mathrm{P}}} r_i^{\mathrm{P}}(x) \to \frac{N_{\mathrm{P}}}{L}\int_{-L/2}^{+L/2} r^{\mathrm{P}}(\mu,x)\,\mathrm{d}\mu \approx \frac{N_{\mathrm{P}}}{L}M_{\mathrm{P}}. \tag{17}$$

## Equal weights form a fixed point

In the following, we will show that equal weights $w^{\mathrm{E}}(\mu) = w_0^{\mathrm{E}}$ and $w^{\mathrm{I}}(\mu') = w_0^{\mathrm{I}}$, $\forall \mu, \mu'$ form a fixed point if $w_0^{\mathrm{I}}$ is chosen such that the output neuron fires at the target rate, $\rho_0$, throughout the arena. With equal weights we get a constant firing rate $r_0^{\mathrm{out}}$,

$$r^{\mathrm{out}}(x) = r_0^{\mathrm{out}} = \left[ w_0^{\mathrm{E}}\sum_i r_i^{\mathrm{E}}(x) - w_0^{\mathrm{I}}\sum_i r_i^{\mathrm{I}}(x)\right]_+, \tag{18}$$

which according to *Equation 17* does not depend on $x$. Furthermore, according to *Equation 14*, $\langle r_i^{\mathrm{P}}(x)\rangle_x$ does not depend on the neuron index $i$. Now the stationarity of the excitatory weight evolution follows from *Equation 7*:

$$\frac{\mathrm{d}w_i^{\mathrm{E}}}{\mathrm{d}t} = \eta_{\mathrm{E}}\left\langle r^{\mathrm{out}}\sum_j r_j^{\mathrm{E}}\left(\delta_{ij} - \frac{w_i^{\mathrm{E}}w_j^{\mathrm{E}}}{\sum_k w_k^{\mathrm{E}2}}\right)\right\rangle_x \tag{19}$$

$$= \eta_{\mathrm{E}}r_0^{\mathrm{out}}\sum_j\left[\langle r_j^{\mathrm{E}}\rangle_x\left(\delta_{ij} - \frac{w_0^{\mathrm{E}2}}{N_{\mathrm{E}}w_0^{\mathrm{E}2}}\right)\right] \tag{20}$$

$$= \frac{r_0^{\mathrm{out}}\eta_{\mathrm{E}}M_{\mathrm{E}}}{L}\sum_{j=1}^{N_{\mathrm{E}}}\left(\delta_{ij} - \frac{1}{N_{\mathrm{E}}}\right) = 0, \tag{21}$$

that is, excitatory weights are stationary for all values of $w_0^{\mathrm{E}}$ and $w_0^{\mathrm{I}}$ (here $\delta_{ij}$ denotes the Kronecker delta which is 1 if $i=j$ and 0 otherwise). This holds for all input functions for which $\langle r_j^{\mathrm{E}}(x)\rangle_x$ is independent of $j$. If $r^{\mathrm{out}} = \rho_0$, it immediately follows from *Equation 6* that $\frac{\mathrm{d}w^{\mathrm{I}}}{\mathrm{d}t} = 0$, so the inhibitory weights are stationary if

$$\rho_0 = \mathbf{w}^{\mathrm{E}}\cdot\mathbf{r}^{\mathrm{E}} - \mathbf{w}^{\mathrm{I}}\cdot\mathbf{r}^{\mathrm{I}} = w_0^{\mathrm{E}}\sum_i r_i^{\mathrm{E}} - w_0^{\mathrm{I}}\sum_j r_j^{\mathrm{I}}, \tag{22}$$

which is fulfilled if

$$w_0^{\mathrm{I}} = \frac{w_0^{\mathrm{E}}\sum_i r_i^{\mathrm{E}} - \rho_0}{\sum_j r_j^{\mathrm{I}}} = \frac{w_0^{\mathrm{E}}N_{\mathrm{E}}M_{\mathrm{E}} - \rho_0}{N_{\mathrm{I}}M_{\mathrm{I}}}. \tag{23}$$

## Linear stability analysis

In the following, we will show that the fixed point of equal weights, the *homogeneous steady state*, is unstable when the spatial tuning of inhibitory inputs is broader than that of the excitatory inputs. In this case, perturbations of the fixed point will grow and one particular spatial frequency will grow fastest. We will show that this spatial frequency predicts the spacing of the resulting periodic pattern (*Figure 1g*).

We perturb the fixed point

$$w^{\mathrm{E}}(\mu) = w_0^{\mathrm{E}} + \delta w^{\mathrm{E}}(\mu), \quad w^{\mathrm{I}}(\mu) = w_0^{\mathrm{I}} + \delta w^{\mathrm{I}}(\mu) \tag{24}$$

and look at the time evolution of the perturbations $\frac{d\delta w^E}{dt}$ and $\frac{d\delta w^I}{dt}$ of the excitatory and inhibitory weights around the fixed point.

Close to the fixed point the output neuron fires around the target rate $\rho_0$. We thus ignore the rectification in *Equation 4*, that is, $r^{\text{out}} = \rho_0 + \delta r^{\text{out}}$, with $\delta r^{\text{out}} = \sum_k \delta w_k^E r_k^E - \sum_{k'} \delta w_{k'}^I r_{k'}^I$.

## Time evolution of perturbations of the inhibitory weights

We start with the time evolution of the inhibitory weight perturbations:

$$\frac{d\delta w_i^I}{dt} = \frac{dw_i^I}{dt} = \eta_I \left\langle \left( r^{\text{out}} - \rho_0 \right) r_i^I \right\rangle_x \tag{25}$$

$$= \eta_I \left\langle \left( \rho_0 + \delta r^{\text{out}} - \rho_0 \right) r_i^I \right\rangle_x \tag{26}$$

$$= \eta_I \left\langle r_i^I \delta r^{\text{out}} \right\rangle_x \tag{27}$$

$$= \eta_I \left\langle r_i^I \left( \sum_k \delta w_k^E r_k^E - \sum_{k'} \delta w_{k'}^I r_{k'}^I \right) \right\rangle_x \tag{28}$$

$$= \eta_I \left( \sum_{k=1}^{N_E} \left\langle r_i^I r_k^E \right\rangle \delta w_k^E - \sum_{k'=1}^{N_I} \left\langle r_i^I r_{k'}^I \right\rangle \delta w_{k'}^I \right), \tag{29}$$

where only the rates $\mathbf{r}^P$ depend on $x$. Intuitively, the first term in *Equation 29* means that the rate of change of the inhibitory weight perturbation of the weight associated with one location depends on the excitatory perturbations of the weights associated with every other location, weighted with the overlap (the cross correlation) of the two associated tuning functions (analogous for inhibitory weight perturbations). In the continuum limit, the sums are:

$$\eta^P \sum_{k=1}^{N_{P'}} \left\langle r_i^P r_k^{P'} \right\rangle_x \delta w_k^{P'} \to \eta^P \frac{N_{P'}}{L} \int_{-L/2}^{+L/2} \left\langle r^P(\mu) r^{P'}(\mu') \right\rangle_x \delta w^{P'}(\mu') \, d\mu' \tag{30}$$

$$= \int_{-L/2}^{+L/2} K^{PP'}(\mu, \mu') \delta w^{P'}(\mu') \, d\mu', \tag{31}$$

where we introduce overlap kernels

$$K^{PP'}(\mu, \mu') := \eta^P \frac{N_{P'}}{L} \left\langle r^P(\mu) r^{P'}(\mu') \right\rangle_x \quad P, P' \in \{E, I\}. \tag{32}$$

The overlap $\left\langle r^P(\mu) r^{P'}(\mu') \right\rangle_x$ depends only on the distance of the Gaussian fields, that is,

$$K^{PP'}(\mu, \mu') = K^{PP'}(\mu - \mu'). \tag{33}$$

Taking $L \to \infty$, the time evolution of the perturbations of the inhibitory weights can thus be written as convolutions:

$$\frac{d\delta w^I(\mu)}{dt} = (K^{IE} * \delta w^E)(\mu) - (K^{II} * \delta w^I)(\mu), \tag{34}$$

where $*$ denotes a convolution.

## Time evolution of perturbations of the excitatory weights

To derive the time evolution of the excitatory weights, we first show that the weight normalization term in *Equation 7*, expressed through the projection operator $P_{ij} = \frac{w_i w_j}{\sum_k w_k^2}$, leads to a term that balances homogeneous weight perturbations and a term that can be neglected in the continuum limit.

Let $P$ be the projection operator responsible for the normalization of the excitatory weights by projecting a weight update onto a vector that is orthogonal to the hypersphere of constant $\sum_{i=1}^{N_{\mathrm{E}}}(w_i^{\mathrm{E}})^2$. We now determine the projection operator around the fixed point (We drop the index 'E' in the following, to improve readability):

$$P_{ij} = \frac{(w_0 + \delta w_i)(w_0 + \delta w_j)}{\sum_k (w_0 + \delta w_k)^2} \equiv P_{ij}(\mathbf{w} + \delta \mathbf{w}). \tag{35}$$

Using Taylor's theorem

$$P_{ij}(\mathbf{w} + \delta \mathbf{w}) = P_{ij}(\mathbf{w}) + \sum_{l=1}^{N} \delta w_l \frac{\mathrm{d}P_{ij}(\mathbf{w})}{\mathrm{d}w_l} + \mathcal{O}(\delta \mathbf{w}^2) \tag{36}$$

and $w_l = w_0 \forall l$, we get

$$P_{ij}(\mathbf{w}) = \frac{w_i w_j}{\sum_k w_k^2} = 1/N, \tag{37}$$

$$\frac{\mathrm{d}P_{ij}(\mathbf{w})}{\mathrm{d}w_l} = \frac{\delta_{il} w_j}{\sum_k w_k^2} + \frac{\delta_{jl} w_i}{\sum_k w_k^2} - \frac{w_i w_j 2 w_l}{(\sum_k w_k^2)^2} = \frac{\delta_{il}}{N w_0} + \frac{\delta_{jl}}{N w_0} - \frac{2}{N^2 w_0}. \tag{38}$$

In summary this gives:

$$P_{ij} = \underbrace{\frac{1}{N_{\mathrm{E}}}}_{\equiv P_0 \propto \mathcal{O}(1)} + \underbrace{\frac{1}{N_{\mathrm{E}} w_0^{\mathrm{E}}} \left( \delta w_i^{\mathrm{E}} + \delta w_j^{\mathrm{E}} - \frac{2 \sum_{l=1}^{N_{\mathrm{E}}} \delta w_l^{\mathrm{E}}}{N_{\mathrm{E}}} \right)}_{\equiv \delta P_{ij} \propto \mathcal{O}(\delta \mathbf{w})} + \mathcal{O}(\delta \mathbf{w}^2). \tag{39}$$

Using the perturbed projection operator *Equation 39* with *Equation 7,* we obtain the time evolution of the excitatory weight perturbation to linear order:

$$\frac{\mathrm{d}\delta w_i^{\mathrm{E}}}{\mathrm{d}t} = \frac{\mathrm{d}w_i^{\mathrm{E}}}{\mathrm{d}t} \tag{40}$$

$$= \eta_{\mathrm{E}} \left\langle r^{\mathrm{out}} \sum_j (\delta_{ij} - P_{ij}) r_j^{\mathrm{E}} \right\rangle_x \tag{41}$$

$$= \eta_{\mathrm{E}} \left\langle (\rho_0 + \delta r^{\mathrm{out}}) \sum_j (\delta_{ij} - P_0 - \delta P_{ij}) r_j^{\mathrm{E}} \right\rangle_x \tag{42}$$

$$= \eta_{\mathrm{E}} \underbrace{\left\langle \rho_0 \sum_j (\delta_{ij} - P_0) r_j^{\mathrm{E}} \right\rangle_x}_{=0, \mathrm{cf. Equation 19}} + \left\langle \delta r^{\mathrm{out}} \sum_j (\delta_{ij} - P_0) r_j^{\mathrm{E}} \right\rangle_x - \left\langle \rho_0 \sum_j \delta P_{ij} r_j^{\mathrm{E}} \right\rangle_x + \mathcal{O}(\delta \mathbf{w}^2) \tag{43}$$

$$= \eta_{\mathrm{E}} \left( \underbrace{\langle r_i^{\mathrm{E}} \delta r^{\mathrm{out}} \rangle_x}_{(1)} - P_0 \underbrace{\left\langle \delta r^{\mathrm{out}} \sum_j r_j^{\mathrm{E}} \right\rangle_x}_{(2)} - \rho_0 \underbrace{\left\langle \sum_j \delta P_{ij} r_j^{\mathrm{E}} \right\rangle_x}_{(3)} \right) + \mathcal{O}(\delta \mathbf{w}^2) \tag{44}$$

Term $(1)$ in *Equation 44* has a similar structure as in the inhibitory case (*Equation 27*), and will lead to analogous convolutions. he second term is given by

$$(2) = \frac{1}{N_{\mathrm{E}}} \left\langle \left( \sum_k r_k^{\mathrm{E}} \delta w_k^{\mathrm{E}} - \sum_{k'} r_{k'}^{\mathrm{I}} \delta w_{k'}^{\mathrm{I}} \right) \sum_j r_j^{\mathrm{E}} \right\rangle_x \tag{45}$$

$$= \frac{M_\text{E}}{L} \left\langle \sum_k r_k^\text{E} \delta w_k^\text{E} - \sum_{k'} r_{k'}^\text{I} \delta w_{k'}^\text{I} \right\rangle_x \tag{46}$$

$$= \frac{M_\text{E}}{L} \left( \sum_k \langle r_k^\text{E} \rangle_x \delta w_k^\text{E} - \sum_{k'} \langle r_{k'}^\text{I} \rangle_x \delta w_{k'}^\text{I} \right) \tag{47}$$

$$= \frac{M_\text{E}}{L^2} \left( M_\text{E} \sum_k \delta w_k^\text{E} - M_\text{I} \sum_{k'} \delta w_{k'}^\text{I} \right) \tag{48}$$

$$\text{cont.limit} \rightarrow \frac{M_\text{E}}{L^3} \left( N_\text{E} M_\text{E} \int_{-L/2}^{+L/2} \delta w^\text{E}(\mu') \, \mathrm{d}\mu' - N_\text{I} M_\text{I} \int_{-L/2}^{+L/2} \delta w^\text{I}(\mu'') \, \mathrm{d}\mu'' \right) \tag{49}$$

and the third term by

$$(3) = \frac{\rho_0}{N_\text{E} w_0^\text{E}} \left\langle \sum_j r_j^\text{E} \left( \delta w_i^\text{E} + \delta w_j^\text{E} - \frac{2}{N_\text{E}} \sum_l \delta w_l^\text{E} \right) \right\rangle_x \tag{50}$$

$$= \frac{\rho_0}{N_\text{E} w_0^\text{E}} \sum_j \langle r_j^\text{E} \rangle_x \left( \delta w_i^\text{E} + \delta w_j^\text{E} - \frac{2}{N_\text{E}} \sum_l \delta w_l^\text{E} \right) \tag{51}$$

$$= \frac{\rho_0 M_\text{E}}{N_\text{E} w_0^\text{E} L} \sum_j \left( \delta w_i^\text{E} + \delta w_j^\text{E} - \frac{2}{N_\text{E}} \sum_l \delta w_l^\text{E} \right) \tag{52}$$

$$= \frac{\rho_0 M_\text{E}}{w_0^\text{E} L} \left( \delta w_i^\text{E} + \frac{1}{N_\text{E}} \sum_j \delta w_j^\text{E} - \frac{2}{N_\text{E}} \sum_l \delta w_l^\text{E} \right) \tag{53}$$

$$= \frac{\rho_0 M_\text{E}}{w_0^\text{E} L} \left( \delta w_i^\text{E} - \frac{1}{N_\text{E}} \sum_j \delta w_j^\text{E} \right) \tag{54}$$

$$\text{cont.limit} \rightarrow \frac{\rho_0 M_\text{E}}{w_0^\text{E} L} \left( \delta w^\text{E}(\mu) - \frac{1}{L} \int_{-L/2}^{+L/2} \delta w^\text{E}(\mu') \, \mathrm{d}\mu' \right) \tag{55}$$

$$= \frac{\rho_0 M_\text{E}}{w_0^\text{E} L} \int_{-L/2}^{+L/2} \mathrm{d}\mu' \delta w^\text{E}(\mu') \left[ \delta(\mu - \mu') - \frac{1}{L} \right], \tag{56}$$

where $\delta(\mu - \mu')$ denotes the Dirac delta function. Together, this leads to the time evolution of the excitatory weight perturbations:

$$\frac{\mathrm{d}\delta w^\text{E}(\mu)}{\mathrm{d}t} = \int_{-L/2}^{+L/2} \mathrm{d}\mu' \delta w^\text{E}(\mu') \left[ K^\text{EE}(\mu - \mu') - \frac{\eta_\text{E} \rho_0 M_\text{E}}{w_0^\text{E} L} \delta(\mu - \mu') \right. \tag{57}$$

$$\left. + \frac{\eta_\text{E} M_\text{E}}{L^2} \left( \frac{\rho_0}{w_0^\text{E}} - \frac{N_\text{E} M_\text{E}}{L} \right) \right] \tag{58}$$

$$-\int_{-L/2}^{+L/2} d\mu'' \delta w^I(\mu'') \left[ K^{EI}(\mu-\mu'') - \frac{\eta_E N_I M_E M_I}{L^3} \right].$$ (59)

We now assume $L \gg \sigma_P$ and write everything as convolutions, also trivial ones:

$$\frac{d\delta w^E(\mu)}{dt} = \left( \left[ K^{EE} - \frac{\eta_E \rho_0 M_E}{w_0^E L}\delta + \frac{\eta_E M_E}{L^2}\left( \frac{\rho_0}{w_0^E} - \frac{N_E M_E}{L} \right) \right] * \delta w^E \right)(\mu)$$
$$- \left( \left[ K^{EI} - \frac{\eta_E N_I M_E M_I}{L^3} \right] * \delta w^I \right)(\mu).$$ (60)

## Decoupling of spatial frequencies

The convolutions in *Equations 34 and 60* show how the excitatory and inhibitory weight perturbations at one location influence the time evolution of weights at every other location. Transforming the system to frequency space leads to a drastic simplification: The time evolution of a perturbation of a particular spatial frequency depends only on the excitatory and inhibitory perturbation of the same spatial frequency, that is, the Fourier components decouple. We define the Fourier transform $f(k) \equiv \mathcal{F}[f(\mu)]$ with wavevector $k$ of a function $f(\mu)$ of location $\mu$ as:

$$f(k) \equiv \int_{-\infty}^{+\infty} f(\mu)e^{-ik\mu} d\mu$$ (61)

and note that

$$\int_{-\infty}^{+\infty} e^{-ik\mu} d\mu = 2\pi\delta(k).$$ (62)

Using the Convolution theorem and the linearity of the Fourier transform we get

$$\frac{d\delta w^E(k)}{dt} = \left[ \frac{\eta_E M_E}{L^2}\left( \frac{\rho_0}{w_0^E} - \frac{N_E M_E}{L} \right)\delta w^E(k) + \frac{\eta_E N_I M_E M_I}{L^3}\delta w^I(k) \right]2\pi\delta(k)$$
$$- \frac{\eta_E \rho_0 M_E}{w_0^E L}\delta w^E(k) + \left[ K^{EE}(k)\delta w^E(k) - K^{EI}(k)\delta w^I(k) \right]$$ (63)

and

$$\frac{d\delta w^I(k)}{dt} = K^{IE}(k)\delta w^E(k) - K^{II}(k)\delta w^I(k).$$ (64)

The $\delta(k)$ term in *Equation 63* balances homogeneous perturbations in such a way that the output neuron would still fire at the target rate, if not for permutations at other frequencies. In the following, we drop this term, because we are not interested in spatially homogeneous perturbations. Moreover, the continuum limit is valid only for high densities: $N_P/L \to \infty$. We can thus drop terms of lower order than $N_P/L$, which eliminates the $\frac{\eta_E \rho_0 M_E}{w_0^E L}$ term. Writing the remaining terms of *Equations 63 and 64* as a matrix leads to:

$$\begin{bmatrix} \dot{\delta w^E} \\ \dot{\delta w^I} \end{bmatrix}(k) = \begin{bmatrix} K^{EE}(k) & -K^{EI}(k) \\ K^{IE}(k) & -K^{II}(k) \end{bmatrix}\begin{bmatrix} \delta w^E \\ \delta w^I \end{bmatrix}(k),$$ (65)

which no longer contains terms from the weight normalization. The characteristic polynomial of the above matrix is:

$$\lambda^2 + \lambda\left( K^{II} - K^{EE} \right) + K^{EI}K^{IE} - K^{EE}K^{II} = 0$$ (66)

The difference, $K^{EI}K^{IE} - K^{EE}K^{II}$, vanishes for Gaussian input, because:

$$K^{PP'}(\mu, \mu' = 0) = \frac{\eta^P N_{P'}}{L}\left\langle r^P(\mu)r^{P'}(0) \right\rangle_x$$ (67)

$$= \frac{\alpha_P \alpha_{P'} \eta^P N_{P'}}{L^2}\int_{-L/2}^{+L/2} dx \exp\left\{ -\frac{(x-\mu)^2}{2\sigma_P^2} - \frac{x^2}{2\sigma_{P'}^2} \right\}$$ (68)

$$\approx \frac{\alpha_P \alpha_{P'} \eta^P N_{P'}}{L^2} \sqrt{\frac{2\pi}{\frac{1}{\sigma_P^2} + \frac{1}{\sigma_{P'}^2}}} \exp\left\{-\frac{\mu^2}{2(\sigma_P^2 + \sigma_{P'}^2)}\right\}, \tag{69}$$

where we completed the square and used $\int_{-\infty}^{+\infty} e^{-ax^2} = \sqrt{\frac{\pi}{a}}$. Taking the Fourier transform and completing the square again gives

$$K^{PP'}(k) = \frac{\eta^P N_{P'} M_P M_{P'}}{L^2} \exp\left\{-\frac{k^2}{2}(\sigma_P^2 + \sigma_{P'}^2)\right\}. \tag{70}$$

and thus $K^{EI}K^{IE} - K^{EE}K^{II} = 0$.

For $P = P'$, *Equation 70* simplifies to:

$$K^{PP}(k) = \frac{\eta^P N_P M_P^2}{L^2} \exp\left\{-k^2\sigma_P^2\right\}. \tag{71}$$

This leads to the eigenvalues:

$$\lambda_0(k) = 0 \tag{72}$$

$$\lambda_1(k) = K^{EE}(k) - K^{II}(k) \tag{73}$$

$$= \frac{1}{L^2}\left(\eta_E M_E^2 N_E \exp\left\{-k^2\sigma_E^2\right\} - \eta_I M_I^2 N_I \exp\left\{-k^2\sigma_I^2\right\}\right), \tag{74}$$

which are shown in *Figure 8a*. Perturbations with spatial frequencies for which $\lambda_1(k)$ is positive will grow. Setting $\frac{d\lambda_1(k)}{dk} = 0$ gives the wavevector $k_{\max}$ of the Fourier component that grows fastest:

$$\frac{2}{L^2}\left(\eta_I M_I^2 N_I \sigma_I^2 k_{\max} \exp\left\{-k_{\max}^2\sigma_I^2\right\} - \eta_E M_E^2 N_E \sigma_E^2 k_{\max} \exp\left\{-k_{\max}^2\sigma_E^2\right\}\right) = 0 \tag{75}$$

$$\Rightarrow \ln(\eta_I M_I^2 N_I \sigma_I^2) - k_{\max}^2\sigma_I^2 = \ln(\eta_E M_E^2 N_E \sigma_E^2) - k_{\max}^2\sigma_E^2 \tag{76}$$

$$\Rightarrow k_{\max} = \sqrt{\frac{\ln\left(\frac{\eta_I M_I^2 N_I \sigma_I^2}{\eta_E M_E^2 N_E \sigma_E^2}\right)}{\sigma_I^2 - \sigma_E^2}}. \tag{77}$$

Assuming that the fastest-growing spatial frequency from the linearized system will prevail, the final spacing of the periodic pattern, $\ell$, is determined by:

$$\ell = 2\pi/k_{\max} = 2\pi\sqrt{\frac{\sigma_I^2 - \sigma_E^2}{\ln\left(\frac{\eta_I M_I^2 N_I \sigma_I^2}{\eta_E M_E^2 N_E \sigma_E^2}\right)}} = 2\pi\sqrt{\frac{\sigma_I^2 - \sigma_E^2}{\ln\left(\frac{\eta_I N_I \alpha_I^2 \sigma_I^4}{\eta_E N_E \alpha_E^2 \sigma_E^4}\right)}}. \tag{78}$$

*Equation 78* is in exact agreement with the grid spacing obtained in simulations (*Figure 1*g). Moreover, it indicates the bifurcation point: When excitation is as smooth as inhibition ($\sigma_E = \sigma_I$), there is no unstable spatial frequency anymore and every perturbation gets balanced (*Figure 1*g compare *Equation 103*). The grid spacing also depends on the ratio of the inhibitory and excitatory parameters $\eta^P, N_P, \alpha_P$ (logarithmic term in *Equation 78*). We confirm this dependence with simulations on the linear track where we increase either $\eta_I$ or $N_I$ or $\alpha_I^2$ such that the product $\gamma = \eta_I N_I \alpha_I^2$ increases with respect to the initial product $\gamma_0$. We find a good agreement with the theoretical prediction for all three variations (*Figure 8b*).

Note that the term $\eta^P M_P^2 N_P$ in the logarithm in *Equation 78* is essentially a factor that determines the rate of weight change of population P: $\eta^P$ is just the scaling factor; $M_P$ is the mass under a tuning function (with quadratic influence: once directly through the firing rate of the input, once through the increased firing rate of the output neuron); $N_P$ is the number of tuning functions. The remaining $\sigma_P^2$ originates specifically from the Gaussian shape of the tuning functions.

## Analysis for non-localized input (Gaussian random fields)

Above, we derived the time evolution of perturbations of excitatory and inhibitory weights for place field-like input, that is, Gaussian tuning curves. In the following we conduct a similar analysis, using non-localized input, that is, random functions with a given spatial autocorrelation length. We show that the grid spacing is predicted by an equation that is equivalent to *Equation 78*.

The non-localized input $r_i^{\mathrm{P}}$ for input neuron $i$ of population $\mathrm{P}$ was obtained by rescaling a Gaussian random field (GRF) $g_i^{\mathrm{P}}$ to mean $1/2$ and minimum 0:

$$r_i^{\mathrm{P}}(x) = \frac{g_i^{\mathrm{P}}(x) - \min_x g_i^{\mathrm{P}}(x)}{2\langle g_i^{\mathrm{P}}(x) - \min_x g_i^{\mathrm{P}}(x)\rangle_x} \ , \tag{79}$$

where $\min_x$ denotes the minimum over all locations and the GRF $g_i^{\mathrm{P}}$ is obtained by convolving a Gaussian $\mathcal{G}^{\mathrm{P}}(x) = \exp(-x^2/2\sigma_{\mathrm{P}}^2)$ with white noise $\xi_i$ from a uniform distribution between $-0.5$ and $0.5$:

$$g_i^{\mathrm{P}}(x) = \int \mathcal{G}^{\mathrm{P}}(x - x')\xi_i^{\mathrm{P}}(x')\, \mathrm{d}x' \ . \tag{80}$$

As the white noise has zero mean, the spatial average of a GRF is also 0 in expectation:

$$\langle g_i^{\mathrm{P}}(x)\rangle_x = \int \langle \mathcal{G}^P(x - x')\rangle_x \xi_i^{\mathrm{P}}(x')\, \mathrm{d}x' \tag{81}$$

$$\propto \int \xi_i^{\mathrm{P}}(x')\, \mathrm{d}x' = 0 \ . \tag{82}$$

The individual minima $\min_x g_i^{\mathrm{P}}(x)$ in *Equation 79* would complicate the subsequent analysis. If we again consider infinitely large systems $L \to \infty$ with infinite density $N_{\mathrm{P}}/L \to \infty$, *Equation 79* simplifies. The mean of the distribution of GRF minima over different input neurons scales logarithmically with the number of samples (*Bovier, 2005*). Here the number of samples corresponds to the number of minima in a GRF, which scales inversely with the width of the convolution kernel that was used to obtain the GRF:

$$\text{Number of minima in a GRF} \propto L/\sigma_{\mathrm{P}} \ . \tag{83}$$

In the continuum limit the variance of the minima distribution over cells decreases and the relative difference between the mean minimum value of excitation and inhibition vanishes (*Figure 8c*):

$$\frac{\log(L/\sigma_{\mathrm{E}}) - \log(L/\sigma_{\mathrm{I}})}{\log(L/\sigma_{\mathrm{E}})} = \frac{\log(\sigma_{\mathrm{I}}/\sigma_{\mathrm{E}})}{\log(L/\sigma_{\mathrm{E}})} \to 0 \ . \tag{84}$$

NB: For the argument it doesn't matter if it scales purely logarithmically or with $\log^{\gamma}$, where $\gamma$ is any exponent.

Thus, we take the minimum value as a constant $m$, which does neither depend on the population nor on the input neuron. This leads to a simplified expression of the input tuning functions:

$$r_i^{\mathrm{P}}(x) = \frac{1}{2}\left(1 - \frac{g_i^{\mathrm{P}}(x)}{m}\right) \ . \tag{85}$$

As $\langle r_i^{\mathrm{P}}\rangle = 0.5$ is independent of $i$, equal excitatory weights are a fixed point for the excitatory learning rule *Equation 7* as described in *Equation 19*. Moreover, the sum over all input neurons does not depend on the location:

$$\sum_{i=1}^{N_{\mathrm{P}}} r_i^{\mathrm{P}}(x) = \frac{1}{2}\left(\sum_{i=1}^{N_{\mathrm{P}}} 1 - \sum_{i=1}^{N_{\mathrm{P}}} g_i^{\mathrm{P}}(x)\right) = \frac{N_{\mathrm{P}}}{2} - \frac{1}{2}\int \mathcal{G}^{\mathrm{P}}(x - x') \underbrace{\sum_{i=1}^{N_{\mathrm{P}}} \xi_i^{\mathrm{P}}(x')}_{=0 \text{ in cont. limit}} \mathrm{d}x' = \frac{N_{\mathrm{P}}}{2} \ . \tag{86}$$

Therefore, given constant excitatory weights, all inhibitory weights can be set to a value $w_0^{\mathrm{I}}$ such that the output neuron fires at the target rate, that is, homogeneous weights are a fixed point of the learning rules, as in the scenario with Gaussian input. Moreover, *Equation 29* holds also for GRF

input. The analysis of the projection operator (see above) of the weight normalization lead to a term of homogeneous weight perturbations and a term that could be neglected in the high density limit. We now omit these terms a priori. The time evolution of excitatory and inhibitory weight perturbations can thus be summarized as (compare *Equations 29 and 44*):

$$\frac{d\delta w_i^P}{dt} = \eta^P \left( \sum_{k=1}^{N_E} \langle r_i^P(x) r_k^E(x) \rangle_x \, \delta w_k^E - \sum_{k'=1}^{N_I} \langle r_i^P(x) r_{k'}^I(x) \rangle_x \, \delta w_{k'}^I \right). \tag{87}$$

The above equation describes the time evolution of each synaptic weight. For the Gaussian input of the earlier sections, each synaptic weight is associated with one location. In the continuum limit we thus identified the synaptic weight associated with location $\mu$ with $w^P(\mu)$. An increase of $w^E(\mu)$ corresponded to an increase in firing at location $\mu$ (and in the surrounding, given by the width of the Gaussian of the excitatory tuning). Analogously, an increase of $w^I(\mu)$ caused a decrease in firing at location $\mu$ (and in the surrounding, given by the width of the Gaussian of the inhibitory tuning). Because of the non-localized tuning of GRF input, each synaptic weight has an influence on the firing rate at many locations. The influence of neuron $i$ of population P at location $\mu$ is expressed by $\xi_i^P(\mu)$. If one wanted to increase the firing rate at a specific location $\mu$ – and not just everywhere – one would thus increase all excitatory weights with high $\xi_i^P(\mu)$ and decrease all excitatory weights with low $\xi_i^P(\mu)$ (note that $\xi^P$ can also be negative). The 'weight' that corresponds to location $\mu$ is thus expressed as:

$$w^P(\mu) := \sum_{i=1}^{N_P} w_i^P \xi_i^P(\mu), \tag{88}$$

where we weight each synaptic weight with the value of the corresponding white noise at location $\mu$. This corresponds to expressing the weights in a basis that is associated with the location and not with the individual input neurons. Combining *Equation 88* and *Equation 87* gives the time evolution of the weight perturbations associated with location $\mu$:

$$\frac{d\delta w^P(\mu)}{dt} = \sum_{i=1}^{N_P} \xi_i^P(\mu) \frac{d\delta w_i^P}{dt} \tag{89}$$

$$= \eta^P \sum_{i=1}^{N_P} \xi_i^P(\mu) \left( \sum_{k=1}^{N_E} \langle r_i^P(x) r_k^E(x) \rangle_x \, \delta w_k^E - \sum_{k'=1}^{N_I} \langle r_i^P(x) r_{k'}^I(x) \rangle_x \, \delta w_{k'}^I \right). \tag{90}$$

We now look at the first term of the above equation, the second term will be treated analogously:

$$\sum_{i=1}^{N_P} \xi_i^P(\mu) \sum_{k=1}^{N_E} \langle r_i^P(x) r_k^E(x) \rangle_x \, \delta w_k^E = \left\langle \left( \sum_{i=1}^{N_P} \xi_i^P(\mu) r_i^P(x) \right) \left( \sum_{k=1}^{N_E} \delta w_k^E r_k^E(x) \right) \right\rangle_x. \tag{91}$$

The sum containing the white noise can be simplified using the zero mean property and the expression for the variance of uniform white noise:

$$\sum_{i=1}^{N_P} \xi_i^P(\mu) r_i^P(x) = \frac{1}{2} \left( \underbrace{\sum_{i=1}^{N_P} \xi_i^P(\mu)}_{=0} - \frac{1}{m} \sum_{i=1}^{N_P} \xi_i^P(\mu) g_i^P(x) \right) \tag{92}$$

$$= -\frac{1}{2m} \sum_{i=1}^{N_P} \int \mathcal{G}^P(x - x') \underbrace{\sum_{i=1}^{N_P} \xi_i^P(\mu) \xi_i^P(x')}_{= \beta N_P \delta(x' - \mu) \text{ in cont. limit}} dx' \tag{93}$$

$$= -\frac{\beta N_P}{2m} \mathcal{G}^P(x - \mu), \tag{94}$$

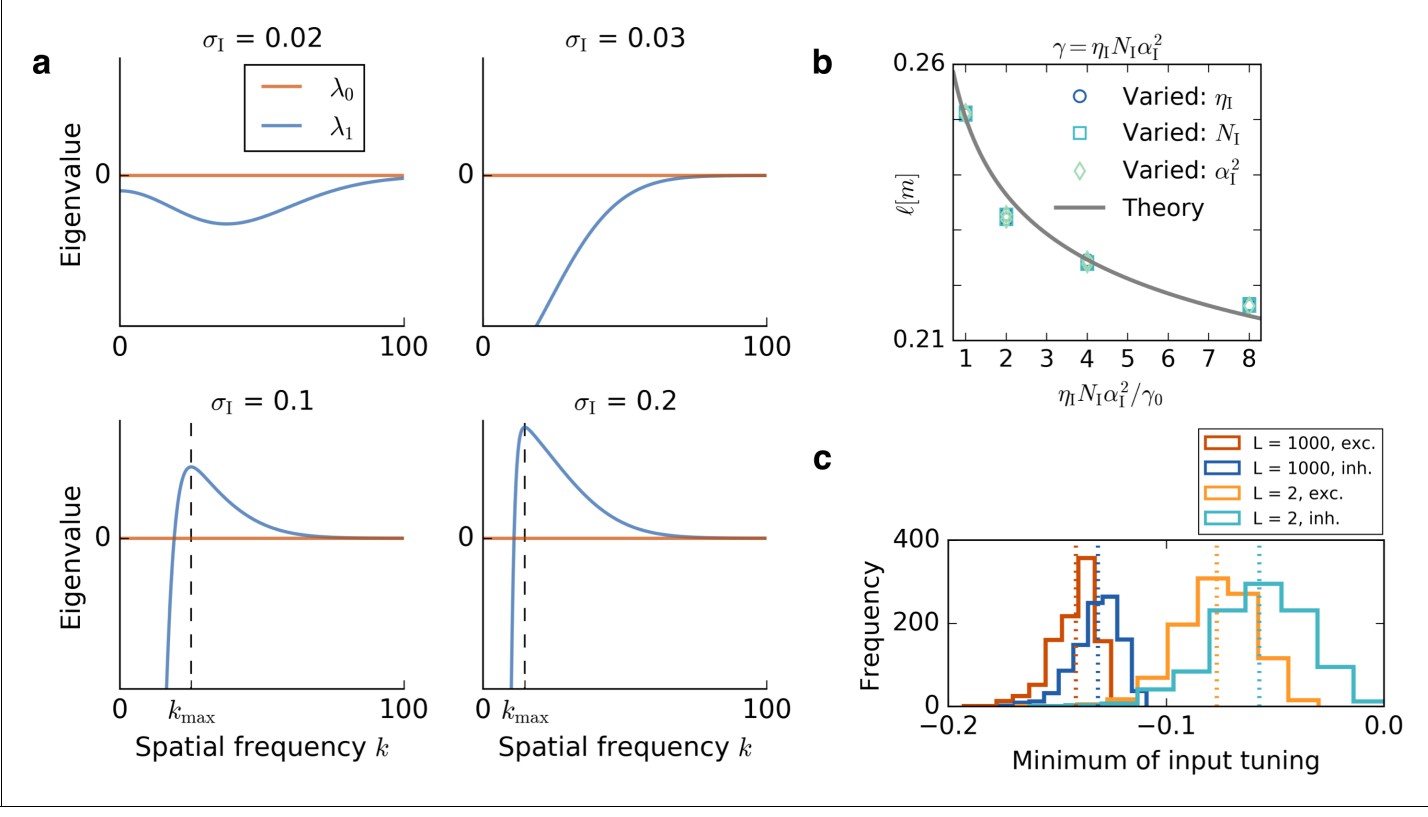

**Figure 8.** Results of the mathematical analysis. (a) The eigenvalue spectrum for the eigenvalues of *Equation 72* for an excitatory tuning of width $\sigma_E = 0.03$. The first eigenvalue $\lambda_0$ is always 0. If the inhibitory tuning is more narrow than the excitatory tuning, that is, $\sigma_I < \sigma_E$, the second eigenvalue $\lambda_1$ is negative for every wavevector $k$. For $\sigma_I > \sigma_E$ the eigenvalue spectrum has a unique positive maximum $k_{max}$, that is, a most unstable spatial frequency. The wavevector $k_{max}$ at which $\lambda_1$ is maximal is obtained from *Equation 78* and marked with a dashed line. (b) The dependence of the grid spacing on learning rate $\eta_I$, number of input neurons $N_I$ and input height $\alpha_I$ is accurately predicted by the theory. The gray line shows the grid spacing obtained from *Equation 78*. We vary the inhibitory learning rate, $\eta_I$ (circles), the number of inhibitory input neurons, $N_I$ (squares), or the square of the height of the inhibitory input place fields, $\alpha_I^2$ (diamonds). The horizontal axis shows the ratio of the product $\eta_I N_I \alpha_I^2$ to the initial value of the product $\gamma_0$. We keep $\eta_E = 0.3 \times 10^{-4}$, $N_E = 800$ and $\alpha_E = 1$ in each simulation and the $\gamma_0$ parameters are: $\eta_I = 0.3 \times 10^{-3}$, $N_I = 200$, $\alpha_I = 1$. (c) Distribution of minimal values of GRF input. Histograms show the distribution of the minimal values of 1000 Gaussian random fields for a small linear track, $L = 2$, and a large linear track $L = 1000$. Red and blue colors correspond to the tuning of excitatory and inhibitory input neurons, respectively. Each dotted line indicates the mean of the histogram of the same color. For larger systems, the distribution of the minimum values gets more narrow and the relative distance between the minima of excitatory and inhibitory neurons decreases.

DOI: https://doi.org/10.7554/eLife.34560.030

where $\beta$ is a proportionality constant that does not depend on the population type P. The Dirac delta $\delta(x' - \mu)$ occurs, because the white noise at different locations is uncorrelated. The sum of the product of weight perturbations and input rates can be rewritten as:

$$\sum_{k=1}^{N_E} \delta w_k^E r_k^E(x) = \frac{1}{2} \left( \underbrace{\sum_{k=1}^{N_E} \delta w_k^E}_{\text{homog. pert.}} - \frac{1}{m} \int \mathcal{G}^E(x - \mu') \underbrace{\sum_{k=1}^{N_E} \delta w_k^E \xi_k^E(\mu')}_{=: \delta w^E(\mu'); \text{ Equation 88}} \, d\mu' \right). \quad (95)$$

The first term is independent of location $x$ and thus will lead only to spatially homogeneous perturbations, which we do not consider in the following. Inserting *Equations 94 and 95* and the analogous terms for inhibition in Equation 91 leads to:

$$\sum_{i=1}^{N_P} \xi_i^P(\mu) \sum_{k=1}^{N_E} \langle r_i^P(x) r_k^E(x) \rangle \delta w_k^E = \frac{\beta N_P}{4m^2} \int \langle \mathcal{G}^P(x - \mu) \mathcal{G}^E(x - \mu') \rangle_x \delta w^E(\mu') \, d\mu' \quad (96)$$

$$= \frac{1}{\eta^{\mathrm{P}}} \int \hat{K}^{\mathrm{PE}}(\mu - \mu') \delta w^{\mathrm{E}}(\mu') \, \mathrm{d}\mu' \tag{97}$$

$$= \frac{1}{\eta^{\mathrm{P}}} (\hat{K}^{\mathrm{PE}} * \delta w^{\mathrm{E}})(\mu), \tag{98}$$

where we introduce kernels for the translation invariant overlap between two Gaussians with different centers (similar to *Equation 32*):

$$\hat{K}^{\mathrm{PP}'}(\mu - \mu') := \frac{\beta \eta \, N_{\mathrm{P}}}{4m^2} \left\langle \mathcal{G}^{\mathrm{P}}(\mu) \mathcal{G}^{\mathrm{P}'}(\mu') \right\rangle_x = \frac{\beta \eta \, N_{\mathrm{P}}}{4m^2} \left\langle \mathcal{G}^{\mathrm{P}}(0) \mathcal{G}^{\mathrm{P}'}(|\mu - \mu'|) \right\rangle_x \tag{99}$$

*Equation 89* can thus be written as:

$$\frac{\mathrm{d}\delta w^{\mathrm{P}}(\mu)}{\mathrm{d}t} = (\hat{K}^{\mathrm{PE}} * \delta w^{\mathrm{E}})(\mu) - (\hat{K}^{\mathrm{PI}} * \delta w^{\mathrm{I}})(\mu), \tag{100}$$

which leads to a dynamical system for the Fourier components of the weight perturbations that is equivalent to *Equation 65* with eigenvalues:

$$\lambda_0(k) = 0 \tag{101}$$

$$\lambda_1(k) = \hat{K}^{\mathrm{EE}}(k) - \hat{K}^{\mathrm{II}}(k) \tag{102}$$

$$= \frac{\beta}{4m^2} \left( \eta_{\mathrm{E}} M_{\mathrm{E}}^2 N_{\mathrm{E}} \exp\{-k^2 \sigma^2\} - \eta_{\mathrm{I}} M_{\mathrm{I}}^2 N_{\mathrm{I}} \exp\{-k^2 \sigma^2\} \right). \tag{103}$$

Thus, we get the same expression for the grid spacing as in the scenario of Gaussian input (with $\alpha_{\mathrm{E}} = \alpha_{\mathrm{I}} = 1$):

$$\ell = \sqrt{\frac{\sigma_{\mathrm{I}}^2 - \sigma_{\mathrm{E}}^2}{\ln\left(\frac{\eta_{\mathrm{I}} \sigma_{\mathrm{I}}^4 N_{\mathrm{I}}}{\eta_{\mathrm{E}} \sigma_{\mathrm{E}}^4 N_{\mathrm{E}}}\right)}}. \tag{104}$$

## Glossary
A summary of notation:

| | |
|---:|:---|
| The rat's position at time $t$ | $\mathbf{x}(t)$ |
| Spatial dimensions $x, y$ and head direction $z$ | $\mathbf{x} = (x, y, z)$ |
| Population label; can be E (excitatory) or I (inhibitory) | P |
| Standard deviation of Gaussian tuning of population P | $\sigma_{\mathrm{P}}$ |
| Spatial autocorrelation length of input of population P | $\sigma_{\mathrm{P,corr}}$ |
| Number of input neurons of population P | $N_{\mathrm{P}}$ |
| Number of place fields per input neuron of population P | $N_{\mathrm{P}}^{\mathrm{f}}$ |
| Firing rate of output neuron | $r^{\mathrm{out}}(\mathbf{x})$ |
| Firing rate of input neuron $i$ of population P | $r_i^{\mathrm{P}}(\mathbf{x})$ |
| Synaptic weight of input neuron $i$ of population P to output neuron | $w_i^{\mathrm{P}}(t)$ |
| Learning rates of excitation and inhibition | $\eta_{\mathrm{E}}, \eta_{\mathrm{I}}$ |
| Target rate of the output neuron | $\rho_0$ |
| Length of linear track | $L$ |
| Height of the Gaussian input fields | $\alpha_{\mathrm{E}}, \alpha_{\mathrm{I}}$ |
| Value of Gaussian with standard deviation $\sigma_{\mathrm{P}}$ at location x | $\mathcal{G}^{\mathrm{P}}(x)$ |
| Von Mises distribution with width $\sigma_{\mathrm{P}}$ that is periodic in $[-L/2, L/2]$ | $\mathcal{M}^{\mathrm{P}}(x)$ |

## Additional information

### Funding

| Funder | Grant reference number | Author |
|---|---|---|
| Bundesministerium für Bildung und Forschung | FKZ 01GQ1201 | Henning Sprekeler |

The funders had no role in study design, data collection and interpretation, or the decision to submit the work for publication.

## Author contributions
Simon Nikolaus Weber, Data curation, Software, Formal analysis, Validation, Investigation, Visualization, Methodology, Writing—original draft, Writing—review and editing; Henning Sprekeler, Conceptualization, Resources, Formal analysis, Supervision, Funding acquisition, Validation, Methodology, Project administration, Writing—review and editing

## Author ORCIDs
Simon Nikolaus Weber http://orcid.org/0000-0002-1169-9879
Henning Sprekeler http://orcid.org/0000-0003-0690-3553

## Decision letter and Author response
Decision letter https://doi.org/10.7554/eLife.34560.036
Author response https://doi.org/10.7554/eLife.34560.037

# Additional files

## Supplementary files
• Transparent reporting form
DOI: https://doi.org/10.7554/eLife.34560.031

## Major datasets
The following previously published dataset was used:

| Author(s) | Year | Dataset title | Dataset URL | Database, license, and accessibility information |
| --- | --- | --- | --- | --- |
| Sargolini F, Fyhn M, Hafting T, McNaughton BL, Witter MP, Moser M, Moser EI | 2006 | Grid cell - raw data | http://www.ntnu.edu/kavli/research/grid-cell-data | Publicly available at the Kavli Institute for Systems Neuroscience |

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

## Appendix 1

DOI: https://doi.org/10.7554/eLife.34560.032

### Rat trajectory

In the linear track model (one dimension, **Figure 1**), we create artificial trajectories $x(t)$. The rat moves along a line of length $L$ with constant velocity $v = 1$ cm per unit time step $\Delta t = 1$. The rat always inverts its direction of motion when it hits either end of the enclosure at $-L/2$ or $L/2$. Additionally, in each unit time step it inverts its direction with a probability of $2v\Delta t/L$, resulting in a typical persistence length of $L/2$.

In the open arena model (two dimensions, **Figures 2**, **3** and **5**), we take trajectories $\mathbf{x}(t)$ from behavioral data (**Sargolini et al., 2006b**) of a rat that moved in a 1 m × 1 m quadratic enclosure. The data provide coherent trajectories in intervals of 10 min. To get a 10-hr trajectory, we concatenate 60 individual trajectories. Different trajectories in our simulations correspond to different random orders of concatenation. A 10-min trajectory contains 30,000 locations. We update the location in every unit time step. A time step thus corresponds to 20 ms. For simulations with a separation wall (**Figure 4**), we use a persistent random walk to constrain the motion of the rat to either side of the arena (see below).

In the model for neurons with head direction tuning (three dimensions, **Figure 6**), we use the same behavioral trajectories as in two dimensions. To account for the experimental observation that the head direction of the animal is only roughly aligned with the direction of motion, we model the head direction as the direction of motion plus a random angle that is drawn in each unit time step from a normal distribution with standard deviation $\pi/6$.

In all dimensions and for the learning rates under consideration, we find that the precise trajectory of the rat has only a small influence on the results (see also **Figure 2—figure supplement 1**).

### Spatial tuning of input neurons

The firing rates of excitatory and inhibitory synaptic inputs $r_i^{\mathrm{E}}$, $r_j^{\mathrm{I}}$ are tuned to the location $x$ of the animal. In the following, we use $x$ and $y$ for the first and second spatial dimensions and $z$ for the head direction. The values of $x, y, z$ are in the range $[-L/2, \ L/2]$. Note that we take the interval of length $L$ even for the dimension of head direction to have spatial and head direction input at the same scale. In the interpretation of head direction input, the periodic interval is to be understood as the full circle of 360 degrees.

We analyzed three different kinds of input tuning functions. Place cells (single Gaussians), several place fields (sum of multiple Gaussians) and non-localized input (Gaussians convolved with white noise). We summarize the tuning functions of neurons from the excitatory and the inhibitory population by referring to them as population P, where $\mathrm{P} \in \{\mathrm{E}, \ \mathrm{I}\}$.

For readability, we define a Gaussian of height 1 with standard deviation $\sigma_{\mathrm{P}}$:

$$\mathcal{G}^{\mathrm{P}}(x) := \exp\left\{-\frac{x^2}{2\sigma^2}\right\}. \tag{105}$$

The input function of the $i$-th neuron of population P with $N_{\mathrm{P}}^{\mathrm{f}}$ place fields per input neuron in one dimension is then given by:

$$r_i^{\mathrm{P}}(x) = \sum_{\beta=1}^{N_{\mathrm{P}}^{\mathrm{f}}} \mathcal{G}^{\mathrm{P}}\left(x - \mu_{i,\beta}^{\mathrm{P}}\right), \tag{106}$$

where $\mu_{i,\beta}^{\mathrm{P}}$ denotes the center location of field number $\beta$ of input neuron $i$ of population P. The scenario of place cell-like inputs is obtained by setting $N_{\mathrm{P}} = 1$.

For higher dimensions we define the center components as $\mu_{i,\beta}^{\mathrm{P}} = (\mu_{i,\beta,x}^{\mathrm{P}}, \mu_{i,\beta,y}^{\mathrm{P}}, \mu_{i,\beta,z}^{\mathrm{P}})$. In two dimensions, the tuning of the $i$-th neuron of population P with $N_{\mathrm{P}}^{\mathrm{f}}$ place fields per input neuron is thus given by:

$$r_i^{\mathrm{P}}(\mathbf{x}) = \sum_{\beta=1}^{N_{\mathrm{P}}^{\mathrm{f}}} \mathcal{G}^{\mathrm{P}}\left(x - \mu_{i,\beta,x}^{\mathrm{P}}\right) \mathcal{G}^{\mathrm{P}}\left(x - \mu_{i,\beta,y}^{\mathrm{P}}\right). \tag{107}$$

Here, the two one-dimensional Gaussians can have different standard deviations along different axes, $\sigma_{\mathrm{P},x}$ and $\sigma_{\mathrm{P},y}$, respectively. For simplicity, we constrain the resulting elliptic bell-shaped curve to be aligned with the $x$ or $y$ axis.

In three dimensions we also consider bell-shaped tuning functions along the $z$-direction. However, as the head direction component is periodic, we take von Mises functions that are periodic in the interval $[-L/2, L/2]$:

$$\mathcal{M}^{\mathrm{P}}(z) := \exp\left\{ \left(\frac{L}{2\pi\sigma_{\mathrm{P},z}}\right)^2 \left[\cos\left(\frac{2\pi z}{L}\right) - 1\right] \right\}. \tag{108}$$

In the interpretation of head direction input, the periodic interval is to be understood as the full circle of 360 degrees. In three dimensions, the tuning of the $i$-th neuron of population P with $N_{\mathrm{P}}^{\mathrm{f}}$ place fields per input neurons is thus given by:

$$r_i^{\mathrm{P}}(\mathbf{x}) = \sum_{\beta=1}^{N_{\mathrm{P}}^{\mathrm{f}}} \mathcal{G}^{\mathrm{P}}\left(x - \mu_{i,\beta,x}^{\mathrm{P}}\right) \mathcal{G}^{\mathrm{P}}\left(y - \mu_{i,\beta,y}^{\mathrm{P}}\right) \mathcal{M}^{\mathrm{P}}\left(z - \mu_{i,\beta,z}^{\mathrm{P}}\right). \tag{109}$$

The center locations $\mu^{\mathrm{p}}$ for neurons of type P in an enclosure of side length $L$ are drawn from a randomly distorted lattice (**Figure 2—figure supplement 5**). First, the total number of input neurons is factorized into its dimensional components $N_P = N_{\mathrm{P},x} N_{\mathrm{P},y} N_{\mathrm{P},z}$. Then, for example along the $x$ dimension, center locations of neurons of population P are placed equidistantly in $[-\frac{L}{2} - 3\sigma_{\mathrm{P},x}, \frac{L}{2} + 3\sigma_{\mathrm{P},x}]$. Allowing the field centers to lie a multiple of their standard deviation outside the box reduces boundary effects. Each point on the equidistant lattice is subsequently distorted with noise drawn from a uniform distribution whose range is given by the distance between two points on the undistorted lattice, that is, $[-\frac{L}{2(N_{\mathrm{P},x}-1)}, \frac{L}{2(N_{\mathrm{P},x}-1)}]$; see **Figure 2—figure supplement 5**. Other dimensions are treated analogously. This procedure ensures a random, but still dense, coverage of the arena with few place fields. A truly random distribution of centers leads to similar results (not shown), but requires more input neurons to cover the arena densely. We create $N_{\mathrm{P}}^{\mathrm{f}}$ of such distorted lattices. To each input neuron we assign one center location from each of the $N_{\mathrm{P}}^{\mathrm{f}}$ lattices at random and without replacement. This guarantees that each input neuron has $N_{\mathrm{P}}^{\mathrm{f}}$ randomly located fields that together cover the arena densely.

We obtain dense non-localized input by convolving Gaussians as in **Equations 105 and 107** (with $N_{\mathrm{P}}^{\mathrm{f}} = 1$) with uniform white noise between $-0.5$ and $0.5$. For the discretization we choose $\sigma_{\mathrm{P}}/20$ and center the Gaussian convolution kernel on an array of eight times its standard deviation. We convolve this array with a sufficiently large array of white noise such that we keep only the values where the array of the convolution kernel is inside the array of the white noise. This way we avoid boundary effects at the edges. From the resulting function we subtract its minimum and then divide by twice the mean of the difference between the function and its minimum. This increases the signal to noise ratio and ensures that all of the inputs have a mean value of 0.5 across the arena and a minimum at 0. For each input neuron we take a different realization of white noise. This results in arbitrary tuning functions of the same autocorrelation length as the Gaussian convolution kernel. We define the autocorrelation length as the distance at which the autocorrelation has decayed to $1/e$ of its maximum, where $e$ is Euler's number. The above mentioned also holds for circular enclosures, but we drop all field centers outside of a circle of radius $L/2 + 3\sigma_{\mathrm{P}}$ because they never get activated. This is not necessary but it reduces simulation time.

## Learning two sides of a room independently

In *Figure 4* we simulated a rat that explores each half of an arena that is divided by a wall. Then the wall is removed and the animal explores the entire arena. This setup was inspired by recent experiments (*Wernle et al., 2018*) and simulations (Rosay et al., in preparation; *Mégevand, 2013*). To simulate the two separated compartments, we use two independent sets of inputs, that is, place cells that are randomly distributed around the entire arena (AB). One set is active when the rat explores the first compartment, the other set is active when the rat explores the second compartment. Both sets are active when the wall is removed. If we used a single set of inputs, the grids would be merged, even before the wall was removed. The excitatory synaptic weights of the two sets are normalized independently. This is important, because otherwise the synaptic weights of inputs that are only active when the rat is in compartment A would die out while the rat explored compartment B.

To constrain the motion of the rat to one side of the arena, we create artificial rat trajectories as a persistent random walk with velocity $v$ along a direction vector $(\cos\phi, \sin\phi)$, with polar angle $\phi$. In each time step, $\Delta t$, a random number drawn from a normal distribution with mean 0 and standard deviation $\sqrt{4v\Delta t/L}$ is added to $\phi$, resulting in a two-dimensional random walk of persistence length $L/2$. Whenever the rat hits one of the boundaries, the direction vector is modified such that the angle of incidence equals the angle of reflection. We relate the trajectory to behavioral times by assuming an average rat velocity of 20 cm/s.

## Boundary effects and stability of grids

The motion of the rat is not periodic. We constrain it to either a square or a circular box. The input tuning is not periodic either. Consequently, input neurons with tuning fields that lie partially outside the boundary receive less activation. This leads to boundary effects: Excitatory weights associated with fields at the boundaries grow less, because the Hebbian learning scales with the presynaptic activation. This leads to smaller firing rates at the boundary. According to the inhibitory learning rule, the inhibitory weights of neurons that are tuned to boundary locations then also grow less. At a distance given by the width of the excitatory firing fields, the excitatory weights grow as fast as those that are far away from the boundary. However, if inhibition is more broadly tuned than excitation, the inhibitory input is still reduced at these locations. Firing fields are thus favored at a distance from the boundary that is determined by the width of the excitatory tuning, because at this location the excitation will exceed the inhibition. This preference of firing at a certain distance from the boundary competes with the preference for hexagonal firing that is induced by the interaction of excitatory and inhibitory plasticity. For place field-like input arranged on a symmetric lattice, the alignment to the boundary can be seen in the alignment of one grid axis to the boundary in a square box (*Figure 2—figure supplement 3a*). This alignment is not an artifact of the symmetric distribution of input fields, because it is not present in a circular arena (*Figure 2—figure supplement 3b*). The tendency to align with the boundary can be overcome using a random distribution of input fields (*Figure 2—figure supplement 3c*), and in particular by using input with more than one place field per neuron, that is, non-localized input. Nonetheless, we observe boundary effects in all simulations when simulating for very long times.

## Distribution of initial synaptic weights

To start with reasonable firing rates, we take the initial weights close to the values that would correspond to the fixed point weights (see also the mathematical analysis). More precisely, initially all synaptic weights are chosen from a uniform distribution. For the spreading of the distribution we take $\pm 5\%$ of the mean value. For the mean value of the excitatory weights, $w_0^{\mathrm{E}}$, we typically take $w_0^{\mathrm{E}} = 1$, see *Table 1*. We then determine the mean of the initial inhibitory weights, $w_0^{\mathrm{I}}$, such that the output neuron fires on average around the target rate:

$$\mathbf{w}^{\mathrm{E}}\mathbf{r}^{\mathrm{E}} - \mathbf{w}^{\mathrm{I}}\mathbf{r}^{\mathrm{I}} = w_0^{\mathrm{E}} \sum_{i=1}^{N_{\mathrm{E}}} r_i^{\mathrm{E}} - w_0^{\mathrm{I}} \sum_{j=1}^{N_{\mathrm{I}}} r_j^{\mathrm{I}} \overset{!}{=} \rho_0 \,, \tag{110}$$

so

$$w_0^{\mathrm{I}} = \frac{w_0^{\mathrm{E}} \sum_{i=1}^{N_{\mathrm{E}}} r_i^{\mathrm{E}} - \rho_0}{\sum_{j=1}^{N_{\mathrm{I}}} r_j^{\mathrm{I}}} \,. \tag{111}$$

The sums are given by:

$$\sum_{i=1}^{N_{\mathrm{P}}} r_i^{\mathrm{P}} = \frac{N_{\mathrm{P}}}{A_{\mathrm{P}}} M_{\mathrm{P}} \,, \tag{112}$$

where $N_{\mathrm{P}}$ is the number of input neurons, $M_{\mathrm{P}}$ is the area under a tuning function and $A_{\mathrm{P}}$ is the area in which the centers of the input tuning function can lie. For the fixed point weight relation *Equation 111* this leads to

$$w_0^{\mathrm{I}} = \frac{w_0^{\mathrm{E}} N_{\mathrm{E}} M_{\mathrm{E}} / A_{\mathrm{E}} - \rho_0}{N_{\mathrm{I}} M_{\mathrm{I}} / A_{\mathrm{I}}} \,. \tag{113}$$

The values for $A_{\mathrm{P}}$ and $M_{\mathrm{P}}$ depend on the dimensionality of the system.

## One dimension

For Gaussian input we have:

$$M_{\mathrm{P}} = \sqrt{2\pi} N_{\mathrm{P}}^{\mathrm{f}} \alpha_{\mathrm{P}} \sigma_{\mathrm{P}}, \quad A_{\mathrm{P}} = L + 6\sigma_{\mathrm{P}} \,. \tag{114}$$

For Gaussian random field input we have:

$$M_{\mathrm{P}} = \frac{A_{\mathrm{P}}}{2}, \quad A_{\mathrm{P}} = L \,. \tag{115}$$

## Two dimensions

For Gaussian input we have:

$$M_{\mathrm{P}} = \int \int r^{\mathrm{P}}(\mu_x, \mu_y) \, \mathrm{d}\mu_x \, \mathrm{d}\mu_y = 2\pi N_{\mathrm{P}}^{\mathrm{f}} \sigma_{\mathrm{P},x} \sigma_{\mathrm{P},y}, \quad A_{\mathrm{P}} = (L + 6\sigma_{\mathrm{P},x})(L + 6\sigma_{\mathrm{P},y}) \,. \tag{116}$$

For Gaussian random field input we have:

$$M_{\mathrm{P}} = \frac{A_{\mathrm{P}}}{2}, \quad A_{\mathrm{P}} = L^2 \,. \tag{117}$$

## Three dimensions

In three dimensions we use a von Mises distribution along the third dimension to account for the periodicity of the head direction angle. We thus get

$$M_{\mathrm{P}} = \int \int \int r^{\mathrm{P}}(\mu_P, \mu_y, \mu_z) \, \mathrm{d}\mu_x \, \mathrm{d}\mu_y \, \mathrm{d}\mu_z \tag{118}$$

$$= N_{\mathrm{P}}^{\mathrm{f}} 2\pi \sigma_{\mathrm{P},x} \sigma_{\mathrm{P},y} L \frac{I_0 \left[ \left( \frac{L}{2\pi \sigma_{\mathrm{P}z}} \right)^2 \right]}{\exp \left\{ \left( \frac{L}{2\pi \sigma_{\mathrm{P}z}} \right)^2 \right\}} \tag{119}$$

where $I_0$ is the modified Bessel function. The area in which the function centers can lie is given by:

$$A_P = (L + \sigma_{P,x})(L + 6\sigma_{P,y})L. \qquad (120)$$

## Grid score measure

We use the grid score suggested by *Langston et al. (2010)* . More precisely, we determine the grid score of a spatial autocorrelogram – the Pearson correlation coefficients for all spatial shifts of the firing rate map against itself – in the following way: We crop a centered doughnut shape from the correlogram. To get the inner radius of the doughnut, we clip all values in the correlogram with values smaller than 0.1 to 0. We obtain the resulting clusters that are larger than 0.1 using scipy.ndimage.measurements.label from the SciPy package for Python with a quadratic filter structure, $((1, 1, 1), (1, 1, 1), (1, 1, 1))$, for a correlogram with $51 \times 51$ pixels. We use the distance from the center to the outermost pixel of the innermost cluster as the inner radius of the doughnut. For the outer radius we try 50 values, linearly increasing from the inner radius to the corner of the quadratic arena. For each of the resulting 50 doughnuts, we rotate the doughnut around the center and correlate it with the unrotated doughnut. We determine the correlation for 30, 60, 90, 120 and 150 degrees. We define the grid score as the minimum of the correlation values at 60 and 120 degrees minus the maximum of the correlation values at 30, 90 and 150 degrees. After trying all 50 doughnuts, we take the highest resulting grid score as the grid score of the cell. A hexagonal symmetry thus leads to positive values, whereas a quadratic symmetry leads to negative values.

## Measure for head direction tuning

To quantify the head direction tuning of a cell, we compare the head direction tuning to a uniform circular tuning, using Watson's $U^2$ measure. We adopted the code from *Mégevand, 2013* . We drew $10{,}000$ samples, s_HD , from a probability distribution created from the head direction tuning array, and $10{,}000$ samples, s_uniform , from a uniform distribution and use watson_u2(s_uniform, s_HD) from *Mégevand, 2013* to quantify the degree of non-circularity. The sharper the head direction tuning, the higher the resulting value.

## Measure for grid spacing on the linear track

We define the grid spacing of one-dimensional grids as the location of the first non-centered peak in the autocorrelogram of the firing pattern (*Figure 1g*). For place cell-like input, we obtain the grid spacing from a single simulation.

For non-localized input the grids show defects, which results in misleading peaks in the correlogram. In this case, we used the first peak of the average of 50 correlograms to get the grid spacing (*Figure 1h*). The 50 correlograms were obtained from 50 realizations that differ only in the randomness of the input function. To avoid taking a fluctuation in the correlogram as the first peak – and thus obtaining misleading grid spacing – we take the maximum between $3\sigma_E$ (to cut out the center of the correlogram) and 1 (a value larger than the largest grid spacing in *Figure 1h*).

For high values of the spatial smoothness of inhibition, $\sigma_I$, the simulation results deviate from the analytical solution. This is because for high $\sigma_I$ but small $\sigma_E$, the output neuron fires very sparsely, which impedes the learning. This can be readily overcome by increasing the tuning width, $\sigma_E$, of the excitatory input.

