## [Decision Letter]

[Editors’ note: a previous version of this study was rejected after peer review, but the authors submitted for reconsideration. The first decision letter after peer review is shown below.]

Thank you for submitting your work entitled "Learning place cells, grid cells and invariances with excitatory and inhibitory plasticity" for consideration by *eLife*. Your article has been reviewed by three peer reviewers, and the evaluation has been overseen by a Senior Editor and a Reviewing Editor. The reviewers have opted to remain anonymous.

The reviewers independently prepare their comments and then engaged in a discussion, coordinated by the Reviewing Editor, leading to the final decision. Based on these discussions and the individual reviews below, we regret to inform you that your work will not be considered further for publication in *eLife*.

While all three reviewers found merit in the work, there were significant questions raised about the generality of the model. In particular, there was the concern that the model relies too heavily on assumed values for certain parameters (inputs, and learning rates), so that the contribution of the model is hard to interpret. For this reason, it was felt that the work, while potentially valuable, would be better directed to a more specialized readership than the general neuroscience readership of *eLife*.

*Reviewer #1:*

The authors present a mathematical model with the ability to generate grid cells, as well as place cells. In this model, each neuron receives excitatory and inhibitory input, which can exhibit varying degrees of smoothness across space. The neural activity evolves according to classic Hebbian plasticity rules, and the relationship between the smoothness of the excitatory versus inhibitory input dictates whether a place field, grid fields, or no spatial tuning at all, is exhibited by the neuron after learning. The authors showed that this model can learn somewhat rapidly, and extended their results to include head direction tuning as well. I think while it is quite bold to present a non-attractor network model of grid cells, especially when there's a lot of evidence that point to grid cells as an attractor network (e.g. Yoon et al., 2013), this model is also interesting and may help explain some of the other non-grid spatial patterns, and conjunctive coding, found in MEC and other parahippocampal areas. The paper is also clear, well-presented, and appears to be very transparent, which is appreciated. However, I think this paper could be further improved by comparing to experimental results, and by fully detailing the caveats and differences with the attractor network model.

1) I don't know if this is possible, but it would be nice if the authors could connect more of the predictions of the model back to data. For example, the authors show that in their conjunctive grid-head direction cells, the head direction preference is also a function of location. The authors state that whether or not this is the case is the data is 'unresolved', but it would be really nice if the authors could dig into this a bit further. This is another prediction that could potentially allow the field to distinguish between this model and the attractor network, and with the amount of grid cell data that is publicly available, it seems possible to at least demonstrate what is currently known.

2) Additionally, it is not completely clear what bells and whistles would have to be added to this model to reproduce the experimental results that the attractor network captures (e.g. that nearby grid cells have a similar orientation and spacing, and may preferentially functionally connect to each other (Dunn et al., 2015)). The authors should expand upon this part so the two models can be compared on a more equal footing.

3) The timescale of learning seems to be a bit of an issue to me. I appreciate that this model does better than others, but I still think that this part of the paper could be expanded upon a bit. When an animal is in a novel environment, the grid pattern appears to be there almost immediately (Hafting et al., 2005). Perhaps this is difficult to actually see, since the animal needs to cover the environment at least once over to see a robust grid pattern. It might be nice to see some examples of what the firing pattern would look like after 3-5 minutes of exploration (perhaps with behavioral trajectories from the Sargolini data), so the model can be mapped directly back to what has been observed experimentally.

Alternatively – I suppose one way around this is that the spatial inputs to the cell don't change with environment, but then this input can't be place cells, which can remap across environments. Perhaps another alternative is that the input comes from non-grid spatial cells in MEC, although these also might not be stable enough across environments (Diehl et al., 2017). Have the authors considered whether the inputs do (or don't) change across environments, and what that might do the stability of the grid pattern?

4) It is a little unclear to me what the spatial inputs are – biologically speaking – to the place cells. The authors state that this is not fully resolved, but I think this should be fleshed out a bit more, given that the entire basis of the model is on these cell types. The input to the grid cells (combinations of place cell, or place cell like cells) seems a bit more grounded, but the authors should expand upon what the inputs to the place cells might actually be. Otherwise, in some parts of the paper, it seems a bit like a place cell is generated from the combined inputs of other place cells (which could also be true, but then as presented feels a bit circular and not as exciting or novel).

5) Is it possible to get non-grid spatial cells, and non-grid spatial cells that also encode head direction and/or running speed, like that seen in Deihl et al., 2017?

*Reviewer #2:*

This paper presents a neural network model of the development of the spatial tunings of different cell types in the hippocampal formation, with special emphasis on grid cells. The authors use rate-based excitatory and inhibitory neurons, in conjunction with simple learning rules, to show that based on the relative smoothness of the spatial profiles of the inhibitory and excitatory inputs, the same learning rule can yield grid-like, place-like, or spatially invariant tunings in the output neurons. In particular, grid cell period is determined by the width of the inhibitory inputs, as shown through simulation and analytics. The authors demonstrate their finding using both simplistic unimodal spatial inputs as well as more realistic non-local, multimodal inputs.

I found the paper to be easy to understand, in particular because the learning rules (Hebbian excitatory and homeostatic Hebbian inhibitory) are straightforward, and their effects on the self-organization of the overall tuning by the rearrangement of activity clusters is intuitive and well described. Further, I found the extension of the model to non-local inputs to be an important result, as too often overly simplistic inputs are used for training. I have a few major comments:

1) As the authors point out, it is well known that grids belonging to the same module share a similar orientation, and that this is an issue for single-cell feedforward models. The authors cite Si et al., 2012 as a potential mechanism for orienting the grids, who show that grid formation and alignment emerge simultaneously. Given that a major thrust of the paper is to show that grids can be learned on the time scale at which they are experimentally ascertained, and therefore that the learning rule is biologically plausible, it is incumbent on the authors to demonstrate that learning to align the grids will not interfere with grid self-organization as proposed in their model, and that it is possible to do so within a reasonable time frame. A supplementary figure should be sufficient.

2) Along similar lines, the authors criticize continuous attractor network (CAN) developmental models of grid cells (in particular, Widloski and Fiete 2014) for their slowness of learning. The authors miss the point here, for while in these models grids are slow to develop during exposure to the first environment, during which the recurrent weights develop (trained on unimodal spatial inputs similar to those used in the author's work), they are rapidly expressed in any other environment afterwards, unfamiliar or not. This is because, once the continuous attractor is established (and this only needs to be done once), grid field expression is simply a matter of network path integration. Thus, grids appear instantly, even in the absence of localized information (e.g., darkness), and shared grid orientation comes for free.

3) In Figure 3—figure supplement 3, the author's state that "early firing fields are still present in the final grid", and say that this is consistent with Hafting et al., 2005. In this figure, the considerable drift in the fields over time makes it hard to relate fields expressed at the end of learning with those at the beginning. I would like to see this quantified in some way other than gridness score, maybe through measuring rate map correlations between the final mature map and the map as it develops in time. To what extent is there phase/scale/orientation drift through time? Further, it is interesting that, according to Figure 3C, the gridness develops rather abruptly, consistent with papers from Wills et al., (Wills et al., 2010, 2012). Does phase/scale/orientation also develop/stabilize with a similar time course?

4) The distinction between feedforward models (for example in this paper) and CAN models is clear, but less so between different feed forward models that self-organize in similar ways. I would like to see these fleshed out a bit more in the Discussion, in particular with regards to Castro and Aguiar, 2014 and Stepanyuk, 2015, as many might not be aware of them.

5) It is well known that grid cells recorded in 1d exhibit tuning curves that are strikingly non-periodic. Recently, it was shown (Yoon et al., 2016) that for cells belonging to the same module, these 1d responses are consistent with slices through the same 2d lattice. The authors propose a model that can potentially develop aperiodic grids when using non-local inputs. However, getting the right aperiodicities across cells so that they correspond to slices through the same 2d lattice would seem difficult.

*Reviewer #3:*

This study proposes a general mechanism for the emergence of the diverse spatial correlates found in the hippocampal formation. Through formal analysis and computational simulations, the authors show that the firing patterns of place cells, grid cells, head direction cells, and related conjunctive correlates can all be obtained by particular forms of synaptic plasticity operating on spatially modulated, excitatory and inhibitory inputs. This framework is interesting for at least two reasons. First, it offers insights that complement those derived from models based on path integration, especially in regard to the sensory-dependent characteristics of these correlates. Second, it suggests a specific role for inhibitory inputs and circuits, which have received increasing attention in the context of the entorhinal cortical architecture and grid cells. Below I summarize my main concerns before providing more detailed comments.

The success of this model appears to rest on the ad hoc design of spatial inputs (both excitatory and inhibitory) that often seem unrealistic, combined with equally ad hoc choices for the learning rates of the synaptic plasticity operating on these inputs (examples discussed below). Moreover, while I appreciate the theoretical appeal of reducing the emergence of the many diverse types of spatial correlates to a single unifying mechanism, this approach is very likely to overlook the functional significance of the complex and varied architecture of the circuits of the hippocampal formation, and to oversimplify the computational interaction of the spatial correlates in these circuits. Thus the proposed framework seems to shift the complexity of the spatial tunings that it aims to explain one synapse upstream, by assuming a perplexing and yet undocumented degree of specificity and organization in the modulation of inhibitory input patterns and their plasticity.

As to the organization of the paper, the mathematical analysis that is described in the Materials and methods section appears an important part of the results in this study. Checking all the mathematical derivations thoroughly is a task better fitting the peer-review style and timeframe of more theoretically oriented journals, and I have only read this part superficially. Still, I think that a summary of the main results of this analysis should be included in the Results section and accompanied by an intuitive explanation. Full derivations should still be provided separately (as they are now, but possibly in an appendix, rather than in the Materials and methods section – again this is because I see the mathematical analysis as one of the contributions/results of this paper, rather than a method). I think such a reorganization would benefit the dissemination of the complete results of this paper to a general audience.

---

## [Author Response]

[Editors’ note: the author responses to the first round of peer review follow.]

Reviewer #1:The authors present a mathematical model with the ability to generate grid cells, as well as place cells. In this model, each neuron receives excitatory and inhibitory input, which can exhibit varying degrees of smoothness across space. The neural activity evolves according to classic Hebbian plasticity rules, and the relationship between the smoothness of the excitatory versus inhibitory input dictates whether a place field, grid fields, or no spatial tuning at all, is exhibited by the neuron after learning. The authors showed that this model can learn somewhat rapidly, and extended their results to include head direction tuning as well. I think while it is quite bold to present a non-attractor network model of grid cells, especially when there's a lot of evidence that point to grid cells as an attractor network (e.g. Yoon et al., 2013), this model is also interesting and may help explain some of the other non-grid spatial patterns, and conjunctive coding, found in MEC and other parahippocampal areas. The paper is also clear, well-presented, and appears to be very transparent, which is appreciated. However, I think this paper could be further improved by comparing to experimental results, and by fully detailing the caveats and differences with the attractor network model.1) I don't know if this is possible, but it would be nice if the authors could connect more of the predictions of the model back to data. For example, the authors show that in their conjunctive grid-head direction cells, the head direction preference is also a function of location. The authors state that whether or not this is the case is the data is 'unresolved', but it would be really nice if the authors could dig into this a bit further. This is another prediction that could potentially allow the field to distinguish between this model and the attractor network, and with the amount of grid cell data that is publicly available, it seems possible to at least demonstrate what is currently known.

We now extended our manuscript to discuss more data. For example we reproduce a recent experiment of coalescing grids in contiguous environments (now Figure 4). We also looked at publicly available data of the head direction tuning of individual grid fields (Figure 6—figure supplement 1, former Figure 5). Unfortunately, this analysis is inconclusive because of a substantial trajectory bias in most of the grid fields, i.e., the distribution of head directions was too uneven in most individual grid fields. This problem tends to be less prominent for grid fields in the center of the arena. A thorough analysis would hence require grid cell recordings with several central firing fields, i.e., smaller grid spacing. Such recordings exist (e.g., Stensola et al. 2015), but we had no success in obtaining these data.

2) Additionally, it is not completely clear what bells and whistles would have to be added to this model to reproduce the experimental results that the attractor network captures (e.g. that nearby grid cells have a similar orientation and spacing, and may preferentially functionally connect to each other (Dunn et al., 2015)). The authors should expand upon this part so the two models can be compared on a more equal footing.

We appreciate that this is the achilles heel of all single cell models. Si et al., 2012 achieved a co-orientation of grid cells using recurrent connectivity, but required a set of intricate mechanisms to simultaneously obtain uncorrelated grid phases. In general, we strongly suspect that obtaining a co-orientation and phase dispersion together – while learning on all synapses – is challenging. In a separate project, a student is investigating to what extent the suggested mechanism can be implemented in a recurrent and fully plastic network. Even on a linear track – hence ignoring orientation of the grid – the phenomenology is very rich and clearly beyond the scope of the present manuscript. CAN models had 10+ years of refinement to accommodate an increasing number of experimental observations, and it is hard to exceed the resulting high bar with a new model.

3) The timescale of learning seems to be a bit of an issue to me. I appreciate that this model does better than others, but I still think that this part of the paper could be expanded upon a bit. When an animal is in a novel environment, the grid pattern appears to be there almost immediately (Hafting et al., 2005). Perhaps this is difficult to actually see, since the animal needs to cover the environment at least once over to see a robust grid pattern. It might be nice to see some examples of what the firing pattern would look like after 3-5 minutes of exploration (perhaps with behavioral trajectories from the Sargolini data), so the model can be mapped directly back to what has been observed experimentally.

The color coding in Figure 3A, B is showing the activity *during learning*, not *after learning*. Thus, the whole learning process is in principle visible in the figure. We now highlight this in the revised figure caption.

Alternatively – I suppose one way around this is that the spatial inputs to the cell don't change with environment, but then this input can't be place cells, which can remap across environments. Perhaps another alternative is that the input comes from non-grid spatial cells in MEC, although these also might not be stable enough across environments (Diehl et al., 2017). Have the authors considered whether the inputs do (or don't) change across environments, and what that might do the stability of the grid pattern?

Thank you very much for this interesting suggestion. In the revised manuscript we studied this and added a small section to the main text and a new figure as a supplement to Figure 3 (Figure 3—figure supplement 2). In short: the output firing pattern is robust to changes of a substantial fraction of the inputs. If all inputs are remapped, a grid is learned anew.

New text in Results paragraph “Rapid appearance of grid cells and their reaction to modifications of the environment”:

“Above, we modeled the exploration of a previously unknown room by assuming the initial synaptic weights to be randomly distributed. […] The strong initial pattern in the weights does not hinder this development (Figure 3—figure supplement 2).”

New text in the Discussion:

“Similarly, we found that room switches in our model lead to grid patterns of the same grid spacing but different phases and orientations. […] It would be interesting to study if a rotation of a fraction of the input would lead to a bimodal distribution of grid rotations: No rotation and co-rotation with the rotated input, as recently observed in experiments where distal cues were rotated but proximal cues stayed fixed (Savelli et al., 2017).”

4) It is a little unclear to me what the spatial inputs are – biologically speaking – to the place cells. The authors state that this is not fully resolved, but I think this should be fleshed out a bit more, given that the entire basis of the model is on these cell types. The input to the grid cells (combinations of place cell, or place cell like cells) seems a bit more grounded, but the authors should expand upon what the inputs to the place cells might actually be. Otherwise, in some parts of the paper, it seems a bit like a place cell is generated from the combined inputs of other place cells (which could also be true, but then as presented feels a bit circular and not as exciting or novel).

The requirement to obtain place cells is that the inhibitory input tuning is very smooth or not tuned to location at all, whereas the excitatory inputs show some kind of tuning to location. The *type* of the excitatory tuning is not crucial, be it place cells, grid cells or some form of non-localized non-periodic cells. In the revised manuscript, we emphasized this finding in the paragraph “Place cells, band cells and stretched grids” and added a simulation to Figure 5 (former Figure 4), where we obtain a place cell from excitatory grid cell input from EC and untuned inhibition.

New text in Results paragraph “Place cells, band cells and stretched grids”:

“The emergence of place cells is independent of the exact shape of the excitatory input. Non-localized inputs (Figure 5A) lead to similar results as grid cell-like inputs of different orientation and grid spacing (Figure 5B, Materials and methods); for other models for the emergence of place cells from grid cells see (Solstad et al., 2006; Franzius et al., 2007b; Rolls et al., 2006; Molter and Yamaguchi, 2008; Ujfalussy et al., 2009; Savelli and Knierim, 2010).”

5) Is it possible to get non-grid spatial cells, and non-grid spatial cells that also encode head direction and/or running speed, like that seen in Deihl et al., 2017?

Cells that would not be classified as grid cells occur naturally in our model. This can be seen, e.g., in the broad distribution of grid scores in Figure 2. Most of these cells show distorted grids, however. It is not difficult to obtain non-grid spatial cells, e.g., by perturbing the smoothness conditions in the input signals locally. Since there are many ways of introducing irregularities, any particular solution would appear arbitrary. For this reason, we did not include simulations in the manuscript. We now cite Diehl 2017 in the section 'Place cells, band cells and stretched grids’, where we introduce the zoo of firing patterns.

We added a small paragraph on running speed modulation to our Discussion:

“In addition, CAN models require that conjunctive (grid x head direction) cells are positively modulated by running speed. […] We expect that in this case, the output neuron would inherit a speed tuning from the input but would otherwise develop similar spatial tuning patterns.”

Reviewer #2:This paper presents a neural network model of the development of the spatial tunings of different cell types in the hippocampal formation, with special emphasis on grid cells. The authors use rate-based excitatory and inhibitory neurons, in conjunction with simple learning rules, to show that based on the relative smoothness of the spatial profiles of the inhibitory and excitatory inputs, the same learning rule can yield grid-like, place-like, or spatially invariant tunings in the output neurons. In particular, grid cell period is determined by the width of the inhibitory inputs, as shown through simulation and analytics. The authors demonstrate their finding using both simplistic unimodal spatial inputs as well as more realistic non-local, multimodal inputs.I found the paper to be easy to understand, in particular because the learning rules (Hebbian excitatory and homeostatic Hebbian inhibitory) are straightforward, and their effects on the self-organization of the overall tuning by the rearrangement of activity clusters is intuitive and well described. Further, I found the extension of the model to non-local inputs to be an important result, as too often overly simplistic inputs are used for training. I have a few major comments:1) As the authors point out, it is well known that grids belonging to the same module share a similar orientation, and that this is an issue for single-cell feedforward models. The authors cite Si et al., 2012 as a potential mechanism for orienting the grids, who show that grid formation and alignment emerge simultaneously. Given that a major thrust of the paper is to show that grids can be learned on the time scale at which they are experimentally ascertained, and therefore that the learning rule is biologically plausible, it is incumbent on the authors to demonstrate that learning to align the grids will not interfere with grid self-organization as proposed in their model, and that it is possible to do so within a reasonable time frame. A supplementary figure should be sufficient.

See reply to reviewer #1, point 2 above.

2) Along similar lines, the authors criticize continuous attractor network (CAN) developmental models of grid cells (in particular, Widloski and Fiete 2014) for their slowness of learning. The authors miss the point here, for while in these models grids are slow to develop during exposure to the first environment, during which the recurrent weights develop (trained on unimodal spatial inputs similar to those used in the author's work), they are rapidly expressed in any other environment afterwards, unfamiliar or not. This is because, once the continuous attractor is established (and this only needs to be done once), grid field expression is simply a matter of network path integration. Thus, grids appear instantly, even in the absence of localized information (e.g., darkness), and shared grid orientation comes for free.

Thanks for spotting this imprecise phrasing. In the revised manuscript, we removed the citation at this particular point in the text and explicitly comment on this matter in more detail in our Discussion:

“Learning the required connectivity in CAN models can take a long time (Widloski and Fiete, 2014). […] The pattern emerges rapidly, but is not instantaneously present (Figure 3—figure supplement 2).”

3) In Figure 3—figure supplement 3, the author's state that "early firing fields are still present in the final grid", and say that this is consistent with Hafting et al., 2005. In this figure, the considerable drift in the fields over time makes it hard to relate fields expressed at the end of learning with those at the beginning. I would like to see this quantified in some way other than gridness score, maybe through measuring rate map correlations between the final mature map and the map as it develops in time. To what extent is there phase/scale/orientation drift through time? Further, it is interesting that, according to Figure 3C, the gridness develops rather abruptly, consistent with papers from Wills et al., (Wills et al., 2010, 2012). Does phase/scale/orientation also develop/stabilize with a similar time course?

The abruptness in Figure 3C is an artifact of the grid score, which is highly sensitive to displaced firing fields. On reasonable time scales (many tens of hours), we observed no grid drift, i.e., phase, spacing and orientation are rather stable. This can be seen, e.g., in the newly added Figure 3—figure supplement 2C (with remapping fraction 0), which shows the correlation coefficient of the grid pattern after 5h of learning with the grid patterns at earlier and later times. The correlation of the grid at 5 hours with the grid at 10 hours is very high considering the small size of the grid fields, suggesting that fields drift only very little once they are learned. The same figure also illustrates the claimed stability of the firing fields that arise early during learning, as the correlation with the grid pattern after 5h shows a very steep rise at learning onset. We refer to this figure also when we talk about grid stability: See response to major comment #3 of reviewer #1.

4) The distinction between feedforward models (for example in this paper) and CAN models is clear, but less so between different feed forward models that self-organize in similar ways. I would like to see these fleshed out a bit more in the Discussion, in particular with regards to Castro and Aguiar, 2014 and Stepanyuk, 2015, as many might not be aware of them.

In the revised version we now highlight the differences to our model when we cite these papers:

“Other models that explain the emergence of grid patterns from place cell input through synaptic depression and potentiation also develop grid cells in realistic times (Castro and Aguiar, 2014; Stepanyuk, 2015). […] How these models generalize to potentially non-localized input is yet to be shown.”

5) It is well known that grid cells recorded in 1d exhibit tuning curves that are strikingly non-periodic. Recently, it was shown (Yoon et al., 2016) that for cells belonging to the same module, these 1d responses are consistent with slices through the same 2d lattice. The authors propose a model that can potentially develop aperiodic grids when using non-local inputs. However, getting the right aperiodicities across cells so that they correspond to slices through the same 2d lattice would seem difficult.

The aperiodicities from general inputs would indeed not have to be consistent with slices through a 2d grid cell. We added a paragraph to our Discussion:

“A recent analysis has shown that the periodic firing of entorhinal cells in rats that move on a linear track can be assessed as slices through a hexagonal grid (Yoon et al., 2016), which arises naturally in a two dimensional CAN. In our model, we would obtain slices through a hexagonal grid if the rat learns the output pattern in two dimensions and afterwards is constrained to move on a linear track that is part of the same arena. If the rat learns the firing pattern on the linear track from scratch, the firing fields would be periodic.”

Reviewer #3:This study proposes a general mechanism for the emergence of the diverse spatial correlates found in the hippocampal formation. Through formal analysis and computational simulations, the authors show that the firing patterns of place cells, grid cells, head direction cells, and related conjunctive correlates can all be obtained by particular forms of synaptic plasticity operating on spatially modulated, excitatory and inhibitory inputs. This framework is interesting for at least two reasons. First, it offers insights that complement those derived from models based on path integration, especially in regard to the sensory-dependent characteristics of these correlates. Second, it suggests a specific role for inhibitory inputs and circuits, which have received increasing attention in the context of the entorhinal cortical architecture and grid cells. Below I summarize my main concerns before providing more detailed comments.The success of this model appears to rest on the ad hoc design of spatial inputs (both excitatory and inhibitory) that often seem unrealistic, combined with equally ad hoc choices for the learning rates of the synaptic plasticity operating on these inputs (examples discussed below). Moreover, while I appreciate the theoretical appeal of reducing the emergence of the many diverse types of spatial correlates to a single unifying mechanism, this approach is very likely to overlook the functional significance of the complex and varied architecture of the circuits of the hippocampal formation, and to oversimplify the computational interaction of the spatial correlates in these circuits. Thus the proposed framework seems to shift the complexity of the spatial tunings that it aims to explain one synapse upstream, by assuming a perplexing and yet undocumented degree of specificity and organization in the modulation of inhibitory input patterns and their plasticity.

We are a bit surprised by the impression of the reviewer that we chose our input patterns ad-hoc, and interpret this as a signal that we presented our case poorly. The core idea of the model is that it is only statistical properties of the input tuning functions that shape the output patterns (their smoothness, i.e., autocorrelation length), and that details of these patterns do not change the results. Of course, any computational model has to commit to a particular input tuning to run analyses. Since we do not know enough about the actual inputs to the various cell types, we resorted to illustrating that the suggested mechanism produces the same results for various different input tunings, including input tuning functions that minimize any further assumptions (the Gaussian random field inputs in Figures 1, 2, 5 are samples from a maximum-entropy distribution for a given autocorrelation function). We were trying to make the point that the requirements of the model are relatively mild and leave significant flexibility regarding the actual input tuning functions, with the exception of the smoothness assumptions, which we regard as testable predictions. To further strengthen this point, we now added simulations with grid cells as inputs, and a simulation with place-field-like excitation and nonlocalized inhibition. If the reviewer has a suggestion how to make this point even clearer, we would appreciate it.

As to the organization of the paper, the mathematical analysis that is described in the Materials and methods section appears an important part of the results in this study. Checking all the mathematical derivations thoroughly is a task better fitting the peer-review style and timeframe of more theoretically oriented journals, and I have only read this part superficially. Still, I think that a summary of the main results of this analysis should be included in the Results section and accompanied by an intuitive explanation. Full derivations should still be provided separately (as they are now, but possibly in an appendix, rather than in the Materials and methods section – again this is because I see the mathematical analysis as one of the contributions/results of this paper, rather than a method). I think such a reorganization would benefit the dissemination of the complete results of this paper to a general audience.

We also regard the mathematical analysis as a strong point and result of the paper. For this reason, we decided to keep it in the Materials and methods section, to make sure it remains part of the main manuscript. Either option is fine for us, and we would leave this decision to the editors.